# NeuroClips: Towards High-fidelity and Smooth fMRI-to-Video Reconstruction

**Zixuan Gong[1], Guangyin Bao[1], Qi Zhang[1,†], Zhongwei Wan[2], Duoqian Miao[1,†], Shoujin Wang[3], Lei Zhu[1], Changwei Wang[4], Rongtao Xu[4], Liang Hu[1], Ke Liu[5], Yu Zhang[1]**

[1]Tongji University
[2]Ohio State University
[3]University of Technology Sydney
[4]Chinese Academy of Sciences
[5]Beijing Anding Hospital
{gongzx,baogy,zhangqi_cs,dqmiao,izy}@tongji.edu.cn

## Abstract

Reconstruction of static visual stimuli from non-invasion brain activity fMRI achieves great success, owning to advanced deep learning models such as CLIP and Stable Diffusion. However, the research on fMRI-to-video reconstruction remains limited since decoding the spatiotemporal perception of continuous visual experiences is formidably challenging. We contend that the key to addressing these challenges lies in accurately decoding both high-level semantics and low-level perception flows, as perceived by the brain in response to video stimuli. To the end, we propose *NeuroClips*, an innovative framework to decode high-fidelity and smooth video from fMRI. *NeuroClips* utilizes a semantics reconstructor to reconstruct video keyframes, guiding semantic accuracy and consistency, and employs a perception reconstructor to capture low-level perceptual details, ensuring video smoothness. During inference, it adopts a pre-trained T2V diffusion model injected with both keyframes and low-level perception flows for video reconstruction. Evaluated on a publicly available fMRI-video dataset, *NeuroClips* achieves smooth high-fidelity video reconstruction of up to 6s at 8FPS, gaining significant improvements over state-of-the-art models in various metrics, e.g., a 128% improvement in SSIM and an 81% improvement in spatiotemporal metrics. Our project is available at https://github.com/gongzix/NeuroClips.

## 1 Introduction

Decoding visual stimuli from neural activity is crucial and prospective to unraveling the intricate mechanisms of the human brain. In the context of non-invasive approaches, visual reconstruction from functional magnetic resonance imaging (fMRI), such as fMRI-to-image reconstruction, shows high fidelity [1, 2, 3, 4], largely benefiting from advanced deep learning models such as CLIP [5, 6] and Stable Diffusion [7]. This convergence of brain science and deep learning presents a promising data-driven learning paradigm to explore a comprehensive understanding of the advanced perceptual and semantic functions of the cerebral cortex. Unfortunately, fMRI-to-video reconstruction still presents significant hurdles that discourage researchers, since decoding the spatiotemporal perception of a continuous flow of scenes, motions, and objects is formidably challenging.

At first glance, fMRI measures blood oxygenation level-dependent (BOLD) signals by snapshotting a few seconds of brain activity, leading to differential temporal resolutions between fMRI (low) and

---

† Corresponding Authors

38th Conference on Neural Information Processing Systems (NeurIPS 2024).

videos (high). The previously advisable solution to address such differential granularity is to perform self-interpolation on fMRI and downsample video frames to pre-align fMRI and videos. Going further, decoding accurate *high-level semantics* and *low-level perception flows* has a more profound impact on the ability to reconstruct high-fidelity videos from brain activity. Early studies before 2022 struggled with achieving satisfactory reconstruction performance, as they failed to acquire precise semantics from powerful (pre-trained) diffusion models. The latest research MinD-Video [8] guides the diffusion model conditioned on visual fMRI features, making an initial attempt to address the semantic issue. However, it lacks a design of low-level visual detailing, so it significantly diverges from the brain's visual system, exhibiting limitations in perceiving continuous low-level visual details.

The brain's reflection of video stimuli is a crucial factor that influences and enlightens the visual decoding of fMRI-to-video reconstruction. Notably, the human brain perceives videos discretely [9, 10] due to the persistence of vision [11, 12, 13, 14, 15] and delayed memory [16]. It is impractical to perceive every video frame, and instead, only keyframes elicit significant responses in the brain's visual system. Reconstructing keyframes from fMRI avoids the issue of differential temporal resolutions between fMRI and videos. Also, the precise semantics and perceptual details in keyframes ensure the high fidelity and smoothness of reconstructed videos, both within and across successive fMRI inputs. Accordingly, we argue that *utilizing keyframe images as anchors for transitional video reconstruction aligns with the brain's cognitive mechanisms and holds greater promise*.

However, relying solely on manipulating fMRI-to-image models, e.g., incorporated with spatiotemporal conditions for successive image reconstruction, to generate keyframes, easily yields suboptimal outcomes. Research dating back to as early as 1960 [17] has shown that the fleeting sequence of images perceived by the retina is hardly discernible during the process of perception. Instead, what emerges is a phenomenal scene or its intriguing features, which can be regarded as non-detailed, low-level images. Initially, the retina captures these low-level perceptions, and subsequently, the higher central nervous system in the brain focuses on and pursues the details, generating high-level images in the cerebral cortex [18]. This process is reflected in the fMRI signal [19, 20, 21, 22, 23, 24]. The video-to-fMRI process naturally incorporates a combination of both low-level and high-level images. Therefore, the reverse fMRI-to-video reconstruction intuitively benefits from taking into account both the high-level semantic features and the low-level perceptual features. Specifically, *it is necessary to decode the low-level perception flows, such as motions and dynamic scenes, from brain activity to complement keyframes, which enhances the reconstruction of high-fidelity frames and produces smooth videos*.

In light of the above discussion, we propose a novel fMRI-to-video reconstruction framework ***NeuroClips*** that introduces two trainable components of *Perception Reconstructor* and *Semantics Reconstructor* for reconstructing low-level perception flows and keyframes, respectively. **1)** Perception Reconstructor introduces *Inception Extension* and *Temporal Upsampling* modules to adaptively align fMRI with video frames, decoding low-level perception flows, i.e., a blurry video. This blurry video ensures the smoothness and consistency of subsequent video reconstruction. **2)** Semantics Reconstructor adopts a diffusion prior and multiple training strategies to concentrate quantities of high-level semantics from various modalities into fMRI embeddings. These fMRI embeddings are mapped to the CLIP image space and decoded to reconstruct high-quality keyframes. **3)** During inference, *NeuroClips* adopts a pre-trained T2V diffusion model injected with keyframes and low-level perception flows for video reconstruction with high fidelity, smoothness, and consistency.

Extensive experiments have validated the superior performance of *NeuroClips*, which is substantially ahead of SOTA baselines in pixel-level metrics and video consistency. *NeuroClips* achieves a 0.219 improvement (128%) in SSIM and a 0.330 improvement (81%) in spatiotemporal metrics and also performs better overall on most video semantic-level metrics. Meanwhile, using multi-fMRI fusion, *NeuroClips* pioneers the exploration of longer video reconstruction up to 6s at 8FPS.

## 2 Related Work

### 2.1 Visual Reconstruction

After its initial exploration [25], static image reconstruction from fMRI has witnessed remarkable success in recent years. Due to the lack of information in fMRI adapted to deep learning models, a path has been gradually explored that aligns fMRI to specific modal representations such as the common image and text modality [26, 27, 28, 29], and CLIP's [5] rich representation of space is

pretty favored. Then, the aligned representations can then be fed into the diffusion model to complete the image generation. Along this path, a large body of literature demonstrates that reconstructing images at both pixel level and semantic level achieves great results [1, 2, 4, 30]. However, the field of fMRI-to-video reconstruction remains largely unexplored. Early studies [31, 32, 33] attempted to reconstruct low-level visual content from fMRI, using the embedded fMRI as conditions to guide GANs or AEs in generating multiple static images. The reconstructed videos contained little to no clear semantics recognizable by humans. Due to the excellent performance of diffusion models, MinD-Video [8] reconstructed 3FPS videos from fMRI. Despite this success, the smoothness and semantic accuracy of these videos remain unsatisfactory, leaving substantial space for improvement.

## 2.2 Video Diffusion Model

Diffusion models for image generation have gained significant attention in research communities recently [34, 35, 36]. DALLE·2 [37] improved text-image generation by leveraging the CLIP [5] joint representation space. Stable Diffusion [7] enhanced generation efficiency by moving the diffusion process to the latent space of VQVAE [38]. To achieve customized generation with trained diffusion models, many works focused on adding extra condition control networks, such as ControlNet [39] and T2I-Adapter [40]. In the realm of video generation, it is typical to extend existing diffusion models with temporal modeling [41, 42, 43]. Animatediff [44] trained a plug-and-play motion module that can be seamlessly integrated into any customized image diffusion model to form a video generator. Stable Video Diffusion [45] fine-tunes pre-trained diffusion models using high-quality video datasets from multiple views to achieve powerful generation.

## 3 Method

The overall framework of *NeuroClips* is illustrated in Figure 1. *NeuroClips* consists of three essential components: **1) Perception Reconstructor** (PR) generates the blurry but continuous rough video from the perceptual level while ensuring consistency between its consecutive frames. **2) Semantics Reconstructor** (SR) reconstructs the high-quality keyframe image from the semantic level. **3) Inference Process** is the fMRI-to-video reconstruction process, which employs a T2V diffusion model and combines the reconstructions from PR and SR to reconstruct the final exquisite video with high fidelity, smoothness, and consistency. Furthermore, *NeuroClips* also pioneers the exploration of **Multi-fMRI Fusion** for longer video reconstruction.

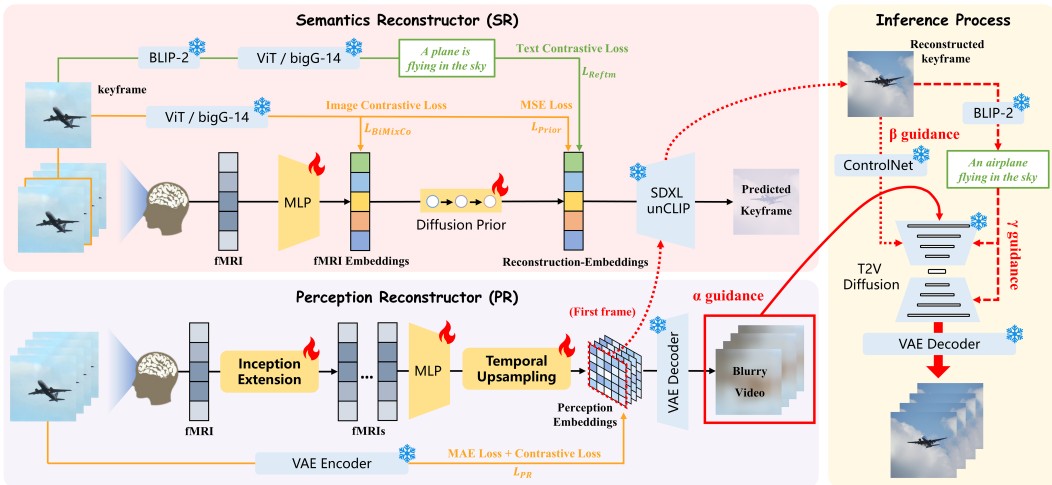

Figure 1: The overall framework of *NeuroClips*. The red lines represent the inernence process.

## 3.1 Perception Reconstructor

Perception reconstruction is essential not only for video reconstruction but also for semantics reconstruction. Additionally, smoothness and consistency are crucial metrics of video quality, and Perception Reconstructor (PR) plays a key role in ensuring these attributes.

We split a video into several clips at two-second intervals (i.e., fMRI time resolution). For each clip $c$, we downsample it and retain part frames at fixed intervals, resulting in a set of frames $\mathbf{X} = [\mathcal{X}_1, \mathcal{X}_2, \cdots, \mathcal{X}_{N_f}]$. $\mathcal{X}_i$ is the $i$-th retained frame image, with a total of $N_f$ retained frames. $\mathcal{Y}_c$ is the corresponding fMRI signal of clip $c$. Here we introduce **Inception Extension** module to extend one fMRI to $N_f$ fMRIs, denoted as $\mathcal{Y}_c \rightarrow \mathbf{Y}$, $\mathbf{Y} = [\mathcal{Y}_1, \mathcal{Y}_2, \cdots, \mathcal{Y}_{N_f}]$.

Sequentially applying a simple MLP and **Temporal Upsampling** module to obtain $\mathbf{Y}$'s embedding set $\mathbf{E}_{\mathcal{Y}} = [e_{\mathcal{Y}_1}, e_{\mathcal{Y}_2}, \cdots, e_{\mathcal{Y}_{N_f}}]$, which can be fed into the Stable Diffusion [7] VAE decoder to produce a series of blurry images. We regard this sequence of blurry images as blurry video. We expect the blurry video to lack semantic content, but to exhibit state-of-the-art perceptual metrics, such as position, shape, scene, etc. Thus, we consider using frame set $\mathbf{X}$ to align $\mathbf{Y}$.

**Training Loss**. Mapping $\mathbf{X}$ to the latent space of Stable Diffusion's VAE to obtain the perception embedding set $\mathbf{E}_{\mathcal{X}} = [e_{\mathcal{X}_1}, e_{\mathcal{X}_2}, \cdots, e_{\mathcal{X}_{N_f}}]$. We adopt mean absolute error (MAE) loss and contrastive loss to train the PR, the overall loss $\mathcal{L}_{PR}$ of PR can be described as:

$$
\begin{aligned}
\mathcal{L}_{PR} = {} & \frac{1}{N_f} \sum_{i=1}^{N_f} |e_{\mathcal{X}_i} - e_{\mathcal{Y}_i}| - \frac{1}{2N_f} \sum_{j=1}^{N_f} \log \frac{\exp(sim(e_{\mathcal{X}_j}, e_{\mathcal{Y}_j})/\tau)}{\sum_{k=1}^{N_f} \exp(sim(e_{\mathcal{X}_j}, e_{\mathcal{Y}_k})/\tau)} \\
& - \frac{1}{2N_f} \sum_{j=1}^{N_f} \log \frac{\exp(sim(e_{\mathcal{Y}_j}, e_{\mathcal{X}_j})/\tau)}{\sum_{k=1}^{N_f} \exp(sim(e_{\mathcal{Y}_j}, e_{\mathcal{X}_k})/\tau)},
\end{aligned}
\tag{1}
$$

where $\tau$ is a temperature hyper-parameter. The function $sim(,)$ is used to compute the similarity.

**Temporal Upsampling**. Due to the low signal-to-noise ratio of fMRI, the direct alignment of fMRI to VAE's pixel space is highly susceptible to overfitting noise, and the learning task is too complex to guarantee the generation of decent blurry images. A common method is aligning to a coarser-grained pixel space and then upsampling to a fine-grained pixel space. The temporal relationship of the frames also needs to be considered in the video task to maintain consistency. Therefore, to achieve consistency between the retained frames, we integrated temporal attention into the upsampling operation. The fMRI embedding $\mathbf{E}_{\mathcal{Y}}$ has five dimensions, i.e. $\mathbf{E}_{\mathcal{Y}} \in \mathbb{R}^{b \times N_f \times c \times h \times w}$, where $b$ denotes the batch size, $c \times h \times w$ is the dimension of embedding space. The upsampling operation merely models spatial relationship, receiving reshaped embedding $\mathbf{E}_{\mathcal{Y}}^{\text{spat}} \in \mathbb{R}^{(b \times N_f) \times c \times h \times w}$ as input. To model temporal relationship of $N_f$ fMRI, we first reshape $\mathbf{E}_{\mathcal{Y}}$ as $\mathbf{E}_{\mathcal{Y}}^{\text{temp}} \in \mathbb{R}^{(b \times h \times w) \times N_f \times c}$. Then, we use learnable mapping to compute the query value $Q = W^Q \mathbf{E}_{\mathcal{Y}}^{\text{temp}}$ and the key value $K = W^K \mathbf{E}_{\mathcal{Y}}^{\text{temp}}$. Finally, the output of temporal attention layer is $\mathbf{E}_{\mathcal{Y}}' = \text{Softmax}(\frac{Q^\top K}{\sqrt{c}}) \cdot \mathbf{E}_{\mathcal{Y}}^{\text{temp}}$. We utilize a learnable mixing coefficient $\eta$ to conduct residual connection:

$$
\mathbf{E}_{\mathcal{Y}} = \eta \cdot \mathbf{E}_{\mathcal{Y}}^{temp} + (1 - \eta) \cdot \mathbf{E}_{\mathcal{Y}}'.
\tag{2}
$$

Based on the above design, PR generates the blurry rough video with initial smoothness and great consistency, laying the foundation for subsequent video reconstruction.

### 3.2 Semantics Reconstructor

Recent cognitive neuroscience studies [46, 47] argue that 'key-frames' play a crucial role in how the human brain recalls and connects relevant memories with unfolding events, and other research [48, 49] also demonstrates that video key-frames can be used as representative features of the entire video clip. Building on these conclusions, the core objective of Semantics Reconstructor (SR) is to reconstruct a high-quality keyframe image that can be used to address the issue of frame rate mismatch between visual stimuli and fMRI signals, thereby enhancing the fidelity of the final exquisite video. The existing fMRI-to-image reconstruction studies [1, 2, 4] facilitate our objective, detailed below:

**fMRI Low-dimensional Processing**. For each clip $c$, its corresponding fMRI signal is $\mathcal{Y}_c$. We use ridge regression to map $\mathcal{Y}_c$ to a lower-dimensional $\mathcal{Y}_c'$ for easier follow-up:

$$
\mathcal{Y}_c' = X(X^T X + \lambda I)^{-1} X^T \mathcal{Y}_c,
\tag{3}
$$

where $X$ is design matrix, $\lambda$ is regularization parameter, and $I$ is identity matrix. Although the human brain processes information in a highly complex and non-linear way, empirical evidence [1, 26, 2]

underscores the effectiveness and sufficiency of linear mapping for achieving desirable reconstruction, due to nonlinear models will easily overfit to fMRI noise and then lead to poor performance [50].

**Alignment of Keyframe Image with fMRI**. Randomly choose one frame in the clip $c$ as its keyframe $\mathcal{X}_c$, and use OpenCLIP ViT-bigG/14 [51] to obtain $e_{\mathcal{X}_c}$, the embedding of keyframe image $\mathcal{X}_c$ in CLIP image space. $e_{\mathcal{Y}_c}$ is the fMRI embedding of $\mathcal{Y}_c'$ via another MLP. Consequently, we perform contrastive learning between $e_{\mathcal{X}_c}$ and $e_{\mathcal{Y}_c}$ to align the keyframe image $\mathcal{X}_c$ and the fMRI $\mathcal{Y}_c$, resulting in enhancing the semantics of $e_{\mathcal{Y}_c}$. It is worth noting that the MLP gets a bidirectional contrastive loss. Previous research [1] has demonstrated that introducing MixCo [52] data augmentation, an extension of mixup utilizing the InfoNCE loss, can effectively help model convergence, especially for scarce fMRI samples. Therefore, the bidirectional loss called BiMixCo $\mathcal{L}_{\text{BiMixCo}}$, which combines MixCo and contrastive loss, needs to be used for training.

**Generation of Reconstruction-Embedding**. Since the embeddings in the CLIP ViT image space are more approximate to real images compared to fMRI embeddings, transforming fMRI embedding $e_{\mathcal{Y}_c}$ into CLIP ViT's image embedding will significantly benefit the reconstruction quality of the keyframe. Therefore, we have to generate the reconstruction-embedding $e_{\mathcal{X}_c}^{re}$ for the keyframe image $\mathcal{X}_c$, essentially, which is the image embedding that will be fed to the subsequent generative model for reconstruction. Inspired by DALLE·2 [37], diffusion prior is an effective approach to transforming embedding. So, we map the fMRI embedding $e_{\mathcal{Y}_c}$ to the OpenCLIP ViT-bigG/14 image space to generate $e_{\mathcal{X}_c}^{re}$. Here, we use the same prior loss $\mathcal{L}_{\text{Prior}}$ in DALLE·2 [37] for training.

**Reconstruction Enhancement from Text Modality**. Original fMRI-to-image reconstruction only relies on visual modality embedding. For instance, reconstructing images conditional on the image embeddings generated by diffusion prior. However, text is another critical modality. Incorporating text with higher semantic density can help improve the semantic content of reconstruction embedding, resulting in making semantics reconstruction more straightforward and effective. We adopt BLIP-2 [53] to introduce the text modality, i.e., the caption $\mathcal{T}_c$ of the keyframe images $\mathcal{X}_c$. Then, we embed $\mathcal{T}_c$ to obtain the text embedding $e_{\mathcal{T}_c}$. Inspired by contrastive learning, we perform contrastive learning between $e_{\mathcal{X}_c}^{re}$ and $e_{\mathcal{T}_c}$ to enhance reconstruction-embedding $e_{\mathcal{X}_c}^{re}$ via additional text modality. The contrastive loss serves as the training loss $\mathcal{L}_{\text{Reftm}}$ of this process, similar to Equation 1, omitted here.

**Training Loss**. As discussed above, the overall training loss $\mathcal{L}_{SR}$ in SR is composite. Therefore, We set mixing coefficients $\delta$ and $\mu$ to balance multiple losses:

$$\mathcal{L}_{SR} = \mathcal{L}_{\text{BiMixCo}} + \delta\mathcal{L}_{\text{Prior}} + \mu\mathcal{L}_{\text{Reftm}}. \tag{4}$$

## 3.3 Inference Process

The inference of *NeuroClips* is the process of fMRI-to-video reconstruction. We jointly utilize the blurry rough video, the high-quality keyframe image, and the additional text modality, which are $\alpha$, $\beta$, and $\gamma$ guidance, to reconstruct the final exquisite video with high fidelity, smoothness, and consistency. And we employ a text-to-video diffusion model to help reconstruct video.

**Text-to-video Diffusion Model**. Pre-training text-to-video (T2V) diffusion models possess a significant amount of prior knowledge from the graphics, image, and video domains. However, like other diffusion models, they face huge challenges in achieving controllable generation. Therefore, directly using the text corresponding to fMRI to reconstruct videos will result in unsatisfactory outcomes, as the semantics of embeddings only originate from the text modality. We also need to enhance the embeddings with semantics from the video and image modalities to produce "composite semantics" embeddings, which aid in achieving controllable generation for the T2V diffusion model.

$\alpha$ **Guidance**. We consider the blurry rough-video $\mathbf{V}_{blurry}$ output from PR as $\alpha$ Guidance. Treating $\mathbf{V}_{blurry}$ as an intermediate noisy video between target video $\mathbf{V}_0$ and noise video $\mathbf{V}_T$, the originally required $T$ steps for the complete forward process can now be reduced to $\vartheta T$ steps. By applying the latent space translation and reparameterization trick, noise $z_T$ can be formalized as:

$$z_T = \sqrt{\bar{\alpha}_T/\bar{\alpha}_{\vartheta T}}\, z_{blurry} + \sqrt{1 - \bar{\alpha}_T/\bar{\alpha}_{\vartheta T}}\, \epsilon, \quad \bar{\alpha}_T = \prod_{t=1}^{T} \alpha_t, \quad \bar{\alpha}_{\vartheta T} = \prod_{t=1}^{\vartheta T} \alpha_t, \tag{5}$$

where $\alpha_t$ represents the noise schedule parameter at time step $t$ and $\epsilon \sim \mathcal{N}(0,1)$ is Gaussian noise. The reverse process involves iteratively denoising the noise video from $T$ steps back to 0 steps.

Adopting a pretrained T2V diffusion model $p_\theta$ to predict the mean and variance of the denoising distribution at each step:

$$z_{t-1} \sim p_\theta(z_{t-1}|z_t) = \mathcal{N}(z_{t-1}; \mu_\theta(z_t, t), \Sigma_\theta(z_t, t)), \tag{6}$$

where $t = T, T - 1, ..., 1$. After translating $z_0$ to pixel space, the reconstructed video $\mathbf{V}_0$ is obtained.

$\beta$ **Guidance**. $\alpha$ Guidance only directs the video generation of the T2V diffusion model at the perception level, leading to significant randomness in the semantics of the reconstructed videos. To resolve this issue, we need to incorporate keyframe images with more supplementary semantics to control the generation process, thereby enhancing the fidelity of the reconstructed videos. Compared to directly reconstructing keyframe images from fMRI embeddings, combining perception embeddings will be more beneficial for maintaining the consistency of structural and semantic information. Therefore, we select the first frame $\mathcal{V}_1$ of blurry $\mathbf{V}_{blurry}$. Input $\mathcal{V}_1$'s embedding and fMRI embedding to SDXL unCLIP [2] (See Appendix D for more discussion about SDXL unCLIP) in SR to reconstruct the keyframe image $\mathcal{X}_{key}$ as $\beta$ Guidance. We employ ControlNet [54] to add $\beta$ Guidance to the T2V diffusion model, in which the keyframes are used as the first-frame to guide video generation.

$\gamma$ **Guidance**. The text is the necessary input for the T2V diffusion model. In order to maintain the consistency of visual semantics, we adopt BLIP-2 [53] to generate the caption for the keyframe image $\mathcal{X}_{key}$, which is used as $\gamma$ Guidance (prompt) for video reconstruction.

The inference process inputs $\alpha$, $\beta$, $\gamma$ Guidance into the T2V diffusion model, and the fMRI-to-video reconstruction can be completed, resulting in the exquisite video with high fidelity and smoothness.

### 3.4 Multi-fMRI Fusion

While it is important to emphasize that single-frame fMRI generates higher frame rate video, the more realistic question is how to recover longer video (longer than fMRI temporal resolution). Previous methods treat single-frame fMRI as a sample, and temporal attention is computed at the single-frame fMRI level, thus failing to generate coherent videos longer than 2s. With the help of *NeuroClips*' SR, we explored the generation of longer videos for the first time. Current video generative models are built on diffusion-based image generation models and attention-based transformer architectures, both of which incur significant computational overhead. As the number of frames increases, the content scales linearly, highlighting the limitations in generating long and complex videos efficiently. Therefore, we chose a more straightforward fusion strategy that does not require additional GPU training. In the inference process, we consider the semantic similarity of two reconstructed keyframes from two neighboring fMRI samples (here we directly determine whether they belong to the same class of objects, e.g., both are jellyfish). Specifically, we obtain the CLIP representations of reconstructed neighboring keyframes and train a shallow MLP based on the representations to distinguish whether two frames share the same class. If semantically similar, we replace the keyframe of the latter fMRI with the tail-frame of the former fMRI's reconstructed video, which will be taken as the first-frame of the latter fMRI to generate the video. As shown in Figure 2, with this strategy, we achieved continuous video reconstruction of up to 6s for the first time.

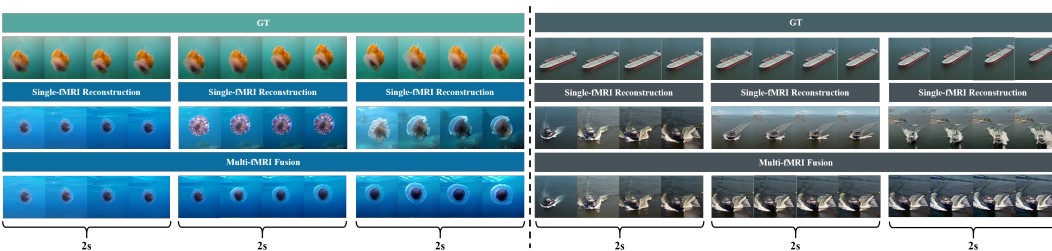

Figure 2: Visualization of Multi-fMRI fusion. With the semantic relevance measure, we can generate video clips up to 6s long without any additional training.

# 4 Experimental Setup

## 4.1 Dataset and Pre-processing

**Dataset.** In this study, we performed fMRI-to-video reconstruction experiments using the open-source fMRI-video dataset (cc2017 dataset[1]) [31]. For each subject, the training and testing video clips were presented 2 and 10 times, respectively, and the testing set was averaged across trials. The dataset consists of a training set containing 18 8-minute video clips and a test set containing 5 8-minute video clips. The MRI (T1 and T2-weighted) and fMRI data (with 2s temporal resolution) were collected using a 3-T MRI system. Thus there are 8640 training samples and 1200 testing samples of fMRI-video pairs.

**Pre-processing.** The fMRI data on the cc2017 were preprocessed using the minimal preprocessing pipeline [55]. The fMRI volumes underwent artifact removal, motion correction (6 DOF), registration to standard space (MNI space), and were further transformed onto the cortical surfaces, which were coregistered onto a cortical surface template [56]. We calculated the voxel-wise correlation between the fMRI voxel signals of each training movie repetition for each subject. The correlation coefficient for each voxel underwent Fisher z-transformation, and the average z scores across 18 training movie segments were tested using the one-sample t-test. The significant voxels (Bonferroni correction, P < 0.05) were considered to be stimulus-activated voxels and used for subsequent analysis A total of 13447, 14828, and 9114 activated voxels were observed in the visual cortex for the three subjects. Following the previous works [3, 57, 58], we used the BOLD signals with a delay of 4 seconds as the movie stimulus responses to account for the hemodynamic delay.

## 4.2 Implementation Details

In this paper, videos from the cc2017 dataset were downsampled from 30FPS to 3FPS to make a fair comparison with the previous methods, and the blurred video was interpolated to 8FPS to generate the final 8FPS video during inference. We split semantic reconstruction into two parts: image contrastive learning and the fine-tuning of the diffusion prior to adapting to the new image distribution space. In PR, all downsampled frames were utilized, and the inception extension was implemented by a shallow MLP. Theoretically, our approach can be used in any text-to-video diffusion model, and we choose the open-source available AnimateDiff [44] as our inference model. $\vartheta$ is set to 0.3 in $\alpha$ Guidance, and the inference is performed with 25 DDIM [35] steps (See Appendix A for more implementation details). All experiments were conducted using a single A100 GPU.

## 4.3 Evaluation Metrics

We conduct the quantitative assessments through frame-based and video-based metrics. Frame-based metrics evaluate each frame individually, providing a snapshot evaluation, whereas video-based metrics evaluate the quality of the video, emphasizing the consistency and smoothness of the video frame sequence. Both are used for a comprehensive analysis from a semantic or pixel perspective.

**Frame-based Metrics** We evaluate frames at the pixel level and semantic level. We use the structural similarity index measure (SSIM) and Peak Signal-to-Noise Ratio (PSNR) as the pixel-level metric and the N-way top-K accuracy classification test (total 1,000 image classes from ImageNet [59]) as the semantics-level metric. To conduct the classification test, we essentially compare the ground truth (GT) classification results with the predicted frame (PF) results using an ImageNet classifier [30, 8]. We consider the trial successful if the GT class is among the top-K probabilities in the PF classification results (We used top-1), selected from N randomly chosen classes that include the GT class. The reported success rate is based on the results of 100 repeated tests.

**Video-based Metrics** We evaluate videos at the semantic level and spatiotemporal(ST)-level. For semantic-level metrics, a similar classification test (total 400 video classes from the Kinetics-400 dataset [60]) is used above, with a video classifier based on VideoMAE [61]. For spatiotemporal-level metrics that measure video consistency, we compute CLIP image embeddings on each frame of the predicted videos and report the average cosine similarity between all pairs of adjacent video frames, which is the common metric CLIP-pcc in video editing.

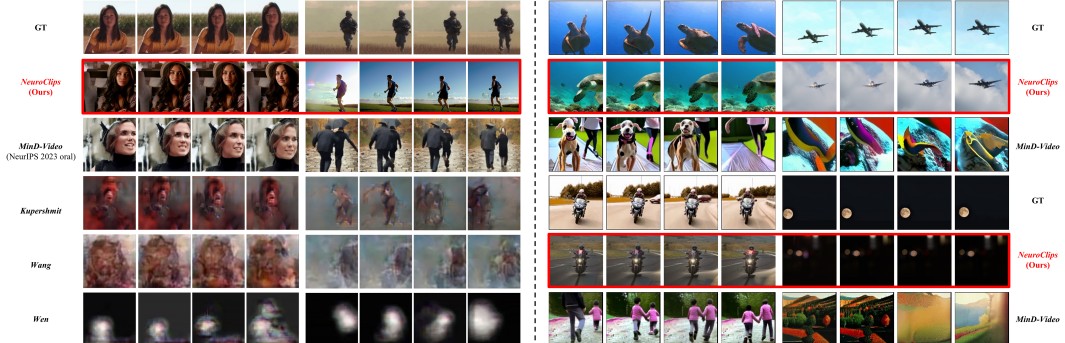

Figure 3: Video reconstruction on the cc2017 dataset. On the left are the results of the comparison with previous studies, and on the right are additional comparisons with previous SOTA methods. Best viewed with zoom-in. As shown in the leftmost figure group, Mind-Video's reconstruction fails to go for detail consistency on the character's face, but our *NeuroClips* achieves an extremely high consistency.

Table 1: Quantitative comparison of *NeuroClips* reconstruction performance against other methods. Bold font signifies the best performance, while underlined text indicates the second-best performance. MinD-Video and *NeuroClips* are both results averaged across all three subjects, and the other methods are results from subject 1. Results of baselines are quoted from [62].

| Method | Video-based | | | Frame-based | | | |
| --- | --- | --- | --- | --- | --- | --- | --- |
| | Semantic-level | | ST-level | Semantic-level | | Pixel-level | |
| | 2-way↑ | 50-way↑ | CLIP-pcc↑ | 2-way↑ | 50-way↑ | SSIM↑ | PSNR↑ |
| Nishimoto [3] | - | - | - | $0.727_{\pm0.04}$ | - | $0.116_{\pm0.09}$ | $8.012_{\pm2.31}$ |
| Wen[31] | - | $0.166_{\pm0.02}$ | - | $0.758_{\pm0.03}$ | $0.070_{\pm0.01}$ | $0.114_{\pm0.15}$ | $7.646_{\pm3.48}$ |
| Wang [32] | $0.773_{\pm0.03}$ | - | $0.402_{\pm0.41}$ | $0.713_{\pm0.04}$ | - | $0.118_{\pm0.08}$ | $\mathbf{11.432}_{\pm2.42}$ |
| Kupershmidt [33] | $0.771_{\pm0.03}$ | - | $0.386_{\pm0.47}$ | $0.764_{\pm0.03}$ | $0.179_{\pm0.02}$ | $0.135_{\pm0.08}$ | $8.761_{\pm2.22}$ |
| MinD-Video [8] | $\mathbf{0.839}_{\pm0.03}$ | $\underline{0.197}_{\pm0.02}$ | $\underline{0.408}_{\pm0.46}$ | $\underline{0.796}_{\pm0.03}$ | $\underline{0.174}_{\pm0.03}$ | $0.171_{\pm0.08}$ | $8.662_{\pm1.52}$ |
| **NeuroClips** | $\underline{0.834}_{\pm0.03}$ | $\mathbf{0.220}_{\pm0.01}$ | $\mathbf{0.738}_{\pm0.17}$ | $\mathbf{0.806}_{\pm0.03}$ | $\mathbf{0.203}_{\pm0.01}$ | $\mathbf{0.390}_{\pm0.08}$ | $\underline{9.211}_{\pm1.46}$ |
| subject 1 | $0.830_{\pm0.03}$ | $0.208_{\pm0.01}$ | $0.736_{\pm0.12}$ | $0.799_{\pm0.03}$ | $0.187_{\pm0.01}$ | $0.392_{\pm0.08}$ | $9.226_{\pm1.42}$ |
| subject 2 | $0.837_{\pm0.03}$ | $0.230_{\pm0.01}$ | $0.742_{\pm0.19}$ | $0.811_{\pm0.03}$ | $0.210_{\pm0.01}$ | $0.392_{\pm0.08}$ | $9.336_{\pm1.52}$ |
| subject 3 | $0.835_{\pm0.03}$ | $0.221_{\pm0.01}$ | $0.735_{\pm0.20}$ | $0.807_{\pm0.03}$ | $0.213_{\pm0.01}$ | $0.387_{\pm0.09}$ | $9.072_{\pm1.44}$ |

## 5 Results

In this section, we compare *NeuroClips* with previous video reconstruction methods on the cc2017 dataset. We provide a visual comparison in Figure 3 and report quantitative metrics in Table 1. We purposely focus on the comparison with the previous SOTA method on the right side of Figure 3 due to the lack of obvious semantic information in the premature method. All methods report results for all test sets except Wen [31], whose results are available for only one segment. Our method generates videos at 8fps and even higher frame rates. For a fair comparison with previous studies, we downsampled the 8FPS videos to 3FPS. Unless otherwise noted, the experimental results and visualizations shown below are all at 3FPS.

As can be seen in Figure 3, earlier methods were unable to produce videos with complete semantics but guaranteed some of the low-level visual features. Compared to MinD-Video, our *NeuroClips* generates single-frame images with higher quality, more precise semantics (e.g., people, turtles, and airplanes), and smoother movements. At the same time, due to the limited data in the training set, some objects in the test set videos did not appear in the training set, such as the moon, and perfect reproduction of the semantics is difficult. However, thanks to our perception reconstructor (aka $\alpha$ Guidance), *NeuroClips* basically reproduces the shape and motion trajectory of the moon, although semantically it is more similar to the aperture, demonstrating the pixel-level reconstruction ability

---

[1] https://purr.purdue.edu/publications/2809/1

of the video. In terms of quantitative metrics, *NeuroClips* significantly outperformed 5 of the 7 metrics, with a 128% improvement in SSIM performance, indicating that the PR complements the lack of pixel-level control. At the semantic level, our metrics overall outperform previous methods, demonstrating the better semantic alignment paradigm of *NeuroClips*. For the ST-level metric, which evaluates video smoothness, *NeuroClips* substantially outperforms MinD-Video because we introduce blurry rough-video (aka $\alpha$ Guidance) for the frozen video generation model, incorporating initial smoothness for video reconstruction. In contrast, MinD-Video lacks perception control, resulting in a substantial decrease in smoothness, as can also be seen in Figure 3, where the human deformations and scene switches are too large within an fMRI frame. In addition, benefiting from our keyframes for first-frame guidance (aka $\beta$ Guidance) and blurry videos, we can connect semantically similar videos to generate longer videos, which may be the reason for the slightly lower video 2-way metrics, as neighboring reconstructed videos are more similar after multi-fMRI fusion.

## 6 Ablations

In this section, we discuss the impact of three important intermediate processes on video reconstruction, including keyframes, blurry videos, and keyframe captioning. The quantitative results are in Table 2 and the visualization results are shown in Figure 4, where all the results of the ablation experiments are from subject 1. Since the keyframe captioning must be obtained through the keyframe, by default we eliminate keyframe and keyframe captioning at the same time. From the quantitative results, it can be seen that *NeuroClips* exhibits better pixel-level metrics without keyframes while showing better semantic-level metrics without blurry clips. This indicates a trade-off between semantic and perception reconstruction, which is similar to the results of a large body of literature on fMRI-to-image reconstruction [1]. The perception recon-

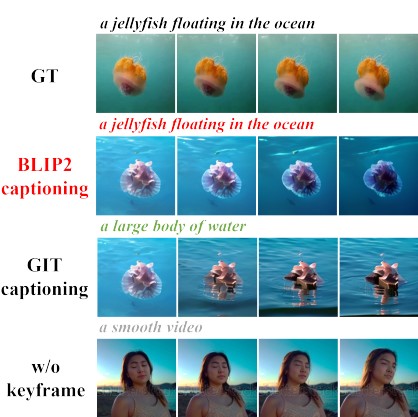

Figure 4: Visualization of ablation study.

struction improves the pixel-level performance and the semantic reconstruction improves the semantic metrics. In addition to this, the best ST-level results for the full model demonstrate the contribution of each module to video consistency.

Table 2: Ablations on the modules of *NeuroClips*, and all results are from subject 1.

| Method | Video-based | | | Frame-based | | | |
| | Semantic-level | | ST-level | Semantic-level | | Pixel-level | |
| | 2-way↑ | 50-way↑ | CLIP-pcc↑ | 2-way↑ | 50-way↑ | SSIM↑ | PSNR↑ |
|---|---|---|---|---|---|---|---|
| w/o keyframe | 0.751±0.04 | 0.164±0.02 | 0.695±0.15 | 0.702±0.04 | 0.128±0.01 | **0.413**±0.09 | **9.334**±1.30 |
| w/o blurry clip | **0.838**±0.03 | **0.213**±0.01 | 0.718±0.11 | **0.805**±0.03 | **0.193**±0.01 | 0.256±0.11 | 8.989±1.37 |
| GIT captioning | 0.828±0.03 | 0.195±0.01 | 0.728±0.12 | 0.785±0.03 | 0.174±0.01 | 0.399±0.08 | 9.297±1.43 |
| **NeuroClips** | 0.830±0.03 | 0.208±0.01 | **0.736**±0.12 | 0.799±0.03 | 0.187±0.01 | 0.392±0.08 | 9.226±1.42 |

For the image captioning method, previous studies [2] have used the GIT [63] model to generate keyframe captions directly from fMRI embeddings in image space, and we generated it from BLIP-2 [53]. Here we compare the text generation results of these two approaches as shown in the Figure 4. We found that GIT's text generation is slightly homogeneous, filled with a large number of similar descriptions, such as '*a large body of water*', '*a man is standing*'. For the diffusion model, the influence of text is significant, so GIT's captions degrade the quality of the semantic reconstruction of the video, e.g., generating flowers on water from jellyfish keyframes. This shows that keyframe-based captioning is more flexible compared to representation-based captioning. Finally, we remove the keyframes and keyframe captions and use only blurred videos to guide the reconstruction, with the text input replaced with the generic description '*a smooth video*'. With this approach, we find that the model generates the video completely blindly, with poor semantic control, demonstrating the strong semantic support that keyframes bring to the *NeuroClips* generation capability. More ablation analysis can be found in Appendix C.4.

# 7 Interpretation Results

To explore the neural interpretability of the model, we visualized voxel-level weights on a brain flat map, where we can see comprehensive structural attention throughout the whole region, as shown in Figure 5. It can be seen that the visual cortex occupies an important position regardless of the task. In addition, for semantic-level reconstruction, the weight distribution of voxels is more spread out over the higher visual cortex, indicating the semantic level of the video. For perceptual reconstruction, the weight distribution of voxels is more concentrated on the lower visual cortex, corresponding to the low-level perception of the human brain. See Appendix C.5 for more subjects' interpretation results.

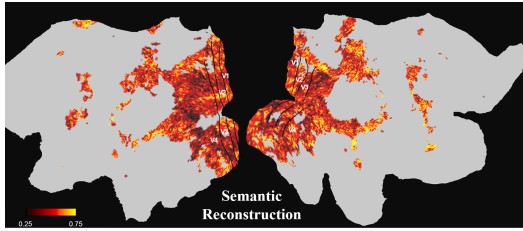 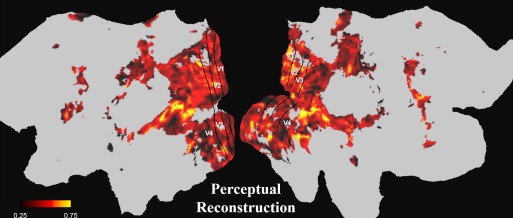

Figure 5: Visualization of voxel weights for the first ridge regression layer for subject 1, with each voxel's weight averaged and normalized to between 0 and 1 and we set the colorbar to 0.25-0.75 for a clear comparison.

# 8 Conclusion

In this paper, we present *NeuroClips*, a novel framework for fMRI-to-video reconstruction. we implement pixel-level and semantic-level visual learning of fMRI through two paths: perception reconstruction and semantic reconstruction. With the learned components, we can configure them into the latest video diffusion models to generate higher quality, higher frame rate, and longer videos without additional training. *NeuroClips* recovers videos with more accurate semantic-level precision and degree of pixel-level matching, establishing a new state-of-the-art in this domain. In addition to this, we visualized the neuroscience interpretability of *NeuroClips* with reliable biological principles.

# 9 Limitations

Although *NeuroClips* has achieved high-fidelity, smooth, and consistent multi-fMRI to video reconstruction, there are still slight flaws. Specifically, our framework is slightly bulky and it relies on extending keyframes to the reconstructed video. A model that can reconstruct videos from the CLIP latent space will avoid this intermediate process. Unfortunately, there is no such available model now. In addition, our method does not reconstruct cross-scene fMRI well, i.e., fMRI recorded during video clip switching. Even if such fMRI scans are in a tiny minority, this will be a direction for future research. Moreover, additional subjects and fMRI recordings should be considered in order to reflect real-world visual experiences sufficiently. However, The alleviation of these limitations will require joint advances in multiple areas and significant further effort will be required. This is because improvements in these areas need to be supported by common developments including machine learning, computer vision, brain science, and biomedicine.

## Ethics Statements

The dataset paper [31] states that informed written consent was obtained from every study participant according to the research protocol approved by the Institutional Review Board at Purdue University.

## Acknowledgements

This research is supported by the National Key Research and Development Program of China (No. 2022YFB3104700), the National Natural Science Foundation of China (No. 61976158, No. 62376198), Shanghai Baiyulan Pujiang Project (No. 08002360429).

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

## A  More Implementation Details

**Training Details.** For Semantic Reconstructor, We first train the fMRI-to-keyframe alignment for 30 epochs with a batch size of 240 and then tune the Diffusion Prior for 150 epochs with a batch size of 64. For Perceptual Reconstructor, we train it for 150 epochs and the batch size is set to 40. We use the AdamW [64] for optimization, with a learning rate set to 3e-4, to which the OneCircle learning rate schedule [65] was set. Mixing coefficients $\delta$ and $\mu$ are set to 30 and 1.

**Inference Process.** In the phase of inference, we use AnimateDiff v3 [44] with Stable Diffusion v1.5 based motion module. The scene model is selected as RealisticVisionV60, and keyframe control is implemented by RGB condition with SparseCtrl [66]. For the video diffusion model, the text prompt is set to the keyframe captioning + ',8k uhd, dslr, soft lighting, high quality, film grain, Fujifilm XT3' and the negative prompt is fixed to 'semi-realistic, cgi, 3d, render, sketch, cartoon, drawing, anime, text, close up, cropped, out of frame, worst quality, low quality, jpeg artifacts, ugly, duplicate, morbid, mutilated, extra fingers, mutated hands, poorly drawn hands, poorly drawn face, mutation, deformed, blurry, dehydrated, bad anatomy, bad proportions, extra limbs, cloned face, disfigured, gross proportions, malformed limbs, missing arms, missing legs, extra arms, extra legs, fused fingers, too many fingers, long neck'.*

**Visualization Software.** We use the Connectome Workbench from Human Connectome Project (HCP), and flatmap templates were selected as 'Q1-Q6_R440.L.flat.32k_fs_LR.surf.gii' and Q1-Q6_R440.R.flat.32k_fs_LR.surf.gii'. Additionally, the cortical parcellation was manually delineated by neuroimaging specialists and neurologists, and aligned with the public templates in FreeSurfer software with verification. We normalized the voxel weights, scaling them to between 0 and 1. Finally, to show a better comparison, the Colorbar was chosen to be 0.25-0.75.

## B  More Details about Method

### B.1  More Details about Perception Reconstructor

We elaborate on the architecture of our Temporal Upsampling in Sec 3.1 here. As illustrated in Figure 6, the Temporal Upsampling module consists of Spatial Layer, Temporal Attention, Learnable Residual Connection, and Upsampling.

The input embedding $\mathbf{E}_{\mathcal{Y}}$ of Temporal Upsampling has five dimensions, i.e. $\mathbf{E}_{\mathcal{Y}} \in \mathbb{R}^{b \times N_f \times c \times h \times w}$ (To express concisely, we have drawn only its $N_f$, $h$, and $w$ in Figure 6.) Before feeding $\mathbf{E}_{\mathcal{Y}}$ into the spatial layer, we reshape it to $\mathbf{E}_{\mathcal{Y}}^{spat} \in \mathbb{R}^{(b \times N_f) \times c \times h \times w}$. First, we conduct modeling in spatial level, using 3D convolution and spatial attention, formalized as $\mathbf{E}_{\mathcal{Y}}' = \mathrm{Spatial}(\mathbf{E}_{\mathcal{Y}}^{spat})$. Note that the spatial layer maintains consistent input and output dimensions so that the dimensions of $\mathbf{E}_{\mathcal{Y}}'$ are exactly the same as those of $\mathbf{E}_{\mathcal{Y}}^{spat}$. Then, applying the learnable residual connection: $\mathbf{E}_{\mathcal{Y}} \leftarrow \eta_1 \cdot \mathbf{E}_{\mathcal{Y}}^{spat} + (1 - \eta_1)\mathbf{E}_{\mathcal{Y}}'$. Subsequently, we reshape $\mathbf{E}_{\mathcal{Y}}$ as $\mathbf{E}_{\mathcal{Y}}^{temp} \in \mathbb{R}^{(b \times h \times w) \times N_f \times c}$ and input $\mathbf{E}_{\mathcal{Y}}^{temp}$ to the temporal layer $\mathbf{E}_{\mathcal{Y}}'' = \mathrm{Temporal}(\mathbf{E}_{\mathcal{Y}}^{temp})$, which is achieved using Temporal Attention as mentioned in main text. Similarly, we apply the learnable residual connection again: $\mathbf{E}_{\mathcal{Y}} \leftarrow \eta_2 \cdot \mathbf{E}_{\mathcal{Y}}^{temp} + (1 - \eta_2)\mathbf{E}_{\mathcal{Y}}''$. Finally, we conduct upsampling to the result $\mathbf{E}_{\mathcal{Y}}$, formalized as $\mathbf{E}_{\mathcal{Y}} \leftarrow \mathrm{Upsampling}(\mathbf{E}_{\mathcal{Y}}) \in \mathbb{R}^{b \times N_f \times c' \times h' \times w'}$. We repeat employing the above module in different $(c, h, w)$ until $\mathbf{E}_{\mathcal{Y}}$ is upsampled to the target dimensions.

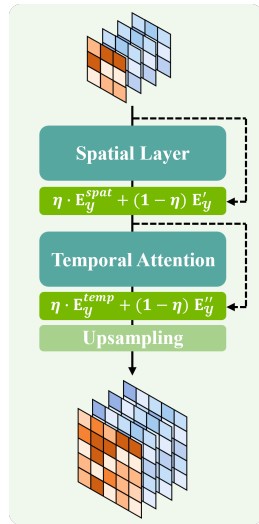

Figure 6: The detailed architecture of Temporal Upsampling module.

### B.2  More Details in Semantic Reconstruction

Here we elaborate on the three loss functions in Equation 4.

**BiMixCo Loss**. BiMixCo aligns the keyframe $\mathcal{X}_c$ and its corresponding fMRI signal $\mathcal{Y}_c$ using bidirectional contrastive loss and MixCo data augmentation. The MixCo needs to mix two independent fMRI signals. For each $\mathcal{Y}_c$, we random sample another fMRI $\mathcal{Y}_{m_c}$, which is the keyframe of the clip index by $m_c$. Then, we mix $\mathcal{Y}_c$ and $\mathcal{Y}_{m_c}$ using a linear combination:

$$\mathcal{Y}_c^* = mix(\mathcal{Y}_c, \mathcal{Y}_{m_c}) = \lambda_c \cdot \mathcal{Y}_c + (1 - \lambda_c)\mathcal{Y}_{m_c}, \tag{7}$$

where $\mathcal{Y}_c^*$ denotes mixed fMRI signal and $\lambda_c$ is a hyper-parameter sampled from Beta distribution. Then, we adapt the ridge regression to map $\mathcal{Y}_c^*$ to a lower-dimensional $\mathcal{Y}_c^{*'}$ and obtain the embedding $e_{\mathcal{Y}_c^*}$ via the MLP, i.e. $e_{\mathcal{Y}_c^*} = \mathcal{E}(\mathcal{Y}_c^{*'})$. Based on this, the BiMixCo loss can be formed as:

$$
\begin{aligned}
\mathcal{L}_{\text{BiMixCo}} = &- \frac{1}{2N_f} \sum_{i=1}^{N_f} \lambda_i \cdot \log \frac{\exp(sim(e_{\mathcal{Y}_i^*}, e_{\mathcal{X}_i})/\tau)}{\sum_{k=1}^{N_f} \exp\left(sim(e_{\mathcal{Y}_i^*}, e_{\mathcal{X}_k})/\tau\right)} \\
&- \frac{1}{2N_f} \sum_{i=1}^{N_f} (1 - \lambda_i) \cdot \log \frac{\exp(sim(e_{\mathcal{Y}_i^*}, e_{\mathcal{X}_{m_i}})/\tau)}{\sum_{k=1}^{N_f} \exp(sim(e_{\mathcal{Y}_i^*}, e_{\mathcal{X}_k})/\tau)} \\
&- \frac{1}{2N_f} \sum_{j=1}^{N_f} \lambda_j \cdot \log \frac{\exp(sim(e_{\mathcal{Y}_j^*}, e_{\mathcal{X}_j})/\tau)}{\sum_{k=1}^{N_f} \exp(sim(e_{\mathcal{Y}_k^*}, e_{\mathcal{X}_j})/\tau)} \\
&- \frac{1}{2N_f} \sum_{j=1}^{N_f} \sum_{\{l|m_l=j\}} (1 - \lambda_j) \cdot \log \frac{\exp(sim(e_{\mathcal{Y}_l^*}, e_{\mathcal{X}_j})/\tau)}{\sum_{k=1}^{N_f} \exp(sim(e_{\mathcal{Y}_k^*}, e_{\mathcal{X}_j})/\tau)},
\end{aligned}
\tag{8}
$$

where $e_{\mathcal{X}_c}$ denotes the OpenCLIP embedding for keyframe $\mathcal{X}_c$.

**Prior Loss**. We use the Diffusion Prior to transform fMRI embedding $e_{\mathcal{Y}_c}$ into the reconstructed OpenCLIP embedding of keyframe $e_{\mathcal{X}_c}^{re}$. Similar to DALLE·2, Diffusion Prior predicts the target embeddings with mean-squared error (MSE) as the supervised objective:

$$\mathcal{L}_{\text{Prior}} = \mathbb{E}_{e_{\mathcal{X}_c}, e_{\mathcal{Y}_c}, \epsilon \sim \mathcal{N}(0,1)} ||\epsilon(e_{\mathcal{Y}_c}) - e_{\mathcal{X}_c}||. \tag{9}$$

**Reftm Loss**. In addition to image representation-level alignment, the assistance of text can aid in generating semantically more compatible images. Considering the inconsistency in the number and dimension of image and text tokens, the previous approaches mostly align the 257 tokens of the image with the CLS token of the text after globally averaging the pooling projection. In this paper, we ignore the CLS token and only map fMRI to 256 tokens of images, so the projection layer of the alignment needs to be adjusted. We develop the Reftm to achieve the alignment between the reconstruction-embedding $e_{\mathcal{X}_c}^{re}$ and its corresponding text embedding $e_{\mathcal{T}_c}$ (the text $\mathcal{T}_c$ is generated by BLIP2 captioning according to the GT keyframe). To align the dimension of reconstruction-embedding $e_{\mathcal{X}_c}^{re}$ and that of text embedding $e_{\mathcal{T}_c}$, we add an adjusted projector $\mathcal{P}(\cdot)$. The adjusted projector directly maps 256 tokens of $e_{\mathcal{X}_c}^{re}$ to the dimension of text embedding. We fine-tuned the projector for 20 epochs on the MSCOCO dataset, which consists of 73k images and 5 texts in each image. And in the training phase of *NeuroClips*, the projector was frozen. Table 3 displays the effect of fine-tuning.

Table 3: The effect of fine-tuning on the MSCOCO 2017.

| Method | Image2Text | Text2Image |
|---|---|---|
| Before fine-tuning | 81.3% | 78.7% |
| After fine-tuning | **95.2%** | **95.2%** |

Finally, for our Semantic Reconstruction, we use CLIP contrastive loss to align reconstruction-embedding $e_{\mathcal{X}_c}^{re}$ and its corresponding text embedding $e_{\mathcal{T}_c}$:

$$\mathcal{L}_{\text{Reftm}} = -\frac{1}{2N_f} \sum_{i=1}^{N_f} \left[ \log \frac{\exp(sim(\mathcal{P}(e_{\mathcal{X}_i}^{re}), e_{\mathcal{T}_i})/\tau)}{\sum_{k=1}^{N_f} \exp(sim(\mathcal{P}(e_{\mathcal{X}_i}^{re}), e_{\mathcal{T}_k})/\tau)} + \log \frac{\exp(sim(\mathcal{P}(e_{\mathcal{X}_i}^{re}), e_{\mathcal{T}_i})/\tau)}{\sum_{k=1}^{N_f} \exp(sim(\mathcal{P}(e_{\mathcal{X}_k}^{re}), e_{\mathcal{T}_i})/\tau)} \right]. \tag{10}$$

# C More Experimental Results

## C.1 Visualizing Reconstructed Keyframe and Text

We display the reconstructed keyframes and their corresponding text in Figure 7. It can be observed that the reconstructed keyframes and ground truth exhibit an extremely high semantic similarity. These results demonstrate the ability of our Semantic Reconstructor to reconstruct semantic-accuracy keyframes and corresponding text descriptions. This reconstructed keyframe contains not only the correct object categories but also detailed information such as object position, color, scene structure, etc., which is crucial for reconstructing high-fidelity videos.

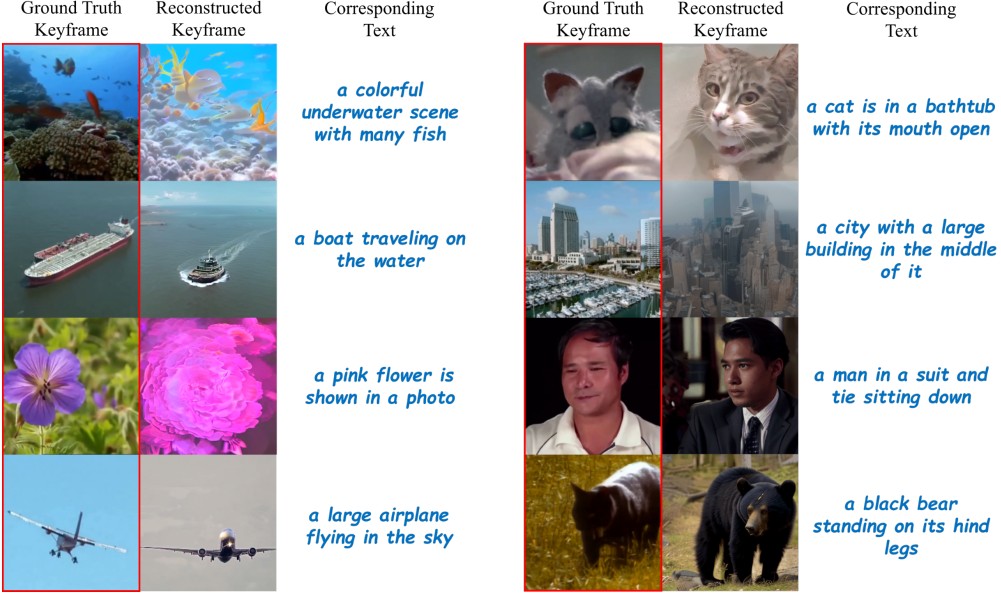

Figure 7: The reconstructed keyframe and its corresponding text. *Best viewed with zoom in.*

## C.2 Visualizing More Successful Reconstructions

We visualized more successfully reconstructed videos in Figure 8. As it can be seen, our *NeuroClips* is capable of successfully reconstructing videos with correct semantics, covering various categories such as portraits, objects, animals, and natural scenes, demonstrating the effectiveness of our Semantic Reconstructor. By conducting detailed comparisons with the ground truth visual stimulus, we find that the structural information and motions in the videos could also be reconstructed, such as the size of the main objects, their positions within the scene and the direction in which a person walks. This demonstrates the critical addition of blurred video to motion and structural information.

We further show videos that are successfully reconstructed by all three subjects in Figure 9, which demonstrated generalization of the model. What's more, we find that these videos often consist of simple backgrounds featuring portraits, animals, or objects, as well as some natural scenes. By comparing the reconstruction results of the three subjects, we discover an interesting phenomenon: the perceived size of objects in simple scenes varies between individuals. For example, the sizes of the jellyfish and airplane reconstructed from different subjects' fMRI show significant differences. These results also suggest, to some extent, the differences in the human visual system among individuals.

## C.3 Visualizing Incorrect Reconstructions

Incorrect results often provide more insights. Therefore, we present some of the incorrect reconstruction results generated by *NeuroClips* in Figure 10. Most incorrect reconstruction clips arise from semantic errors, leading to confusion between semantically similar items, such as fish and turtle, man and woman, and cat and dog. This type of error may arise from two main reasons. On the one hand, the semantic accuracy of the keyframes we reconstructed may be insufficient. We attempt to use

fMRI embedding to retrieve from a pool of keyframe image representations. When the size of the retrieval pool is set to 300, the retrieval accuracy on the test set is approximately 0.22, indicating that there is still significant improvement room for visual semantics encapsulated by the fMRI embedding. On the other hand, the test set may encounter out-of-distribution issues relative to the training set, where semantic categories present in the test set are not seen in the training set, making semantic reconstruction challenging to generalize on the test set.

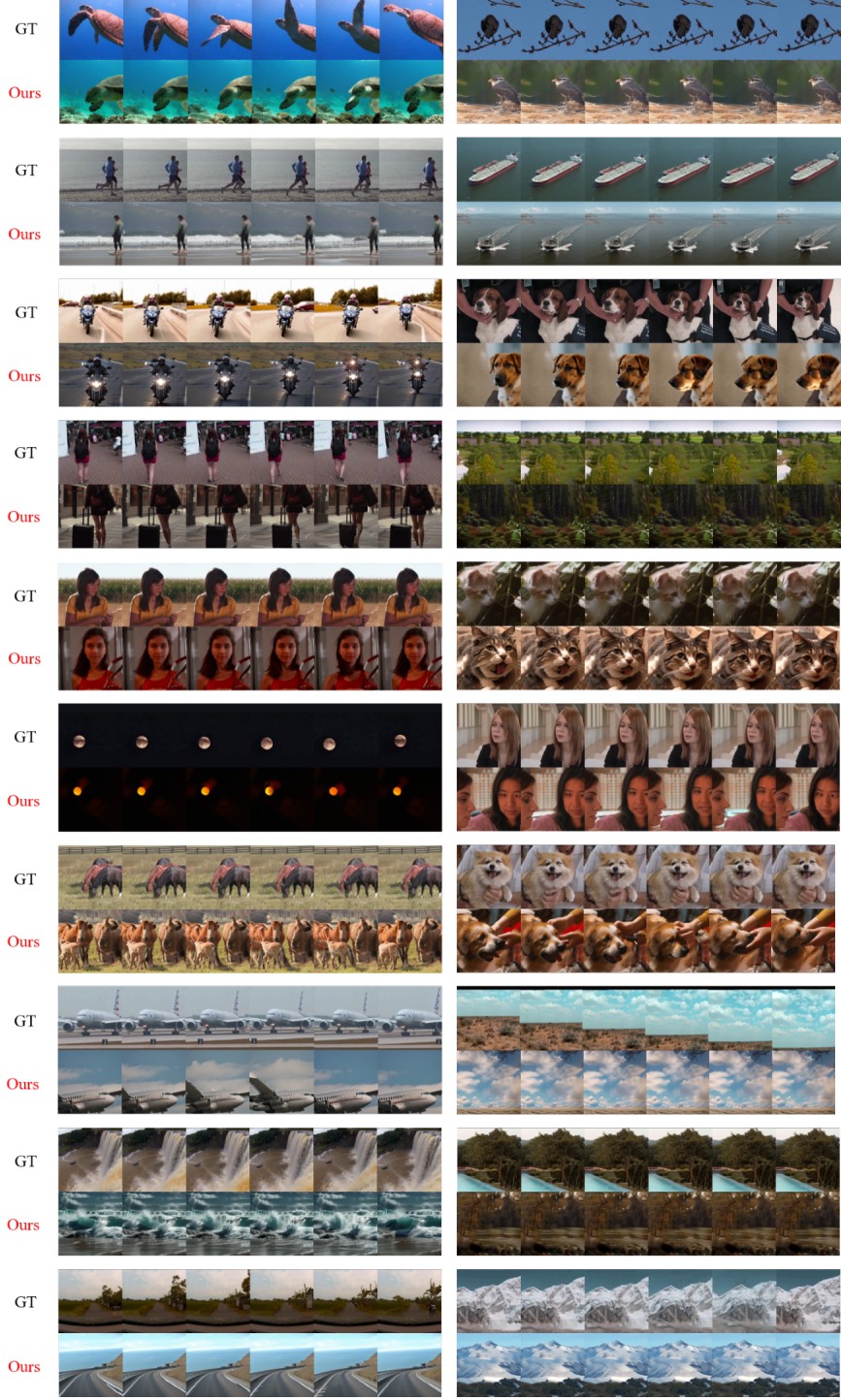

Figure 8: Visualization of more successful reconstruction results. We displayed 6 frames from each 16-frame video generated by fMRI. *Best viewed with zoom in.*

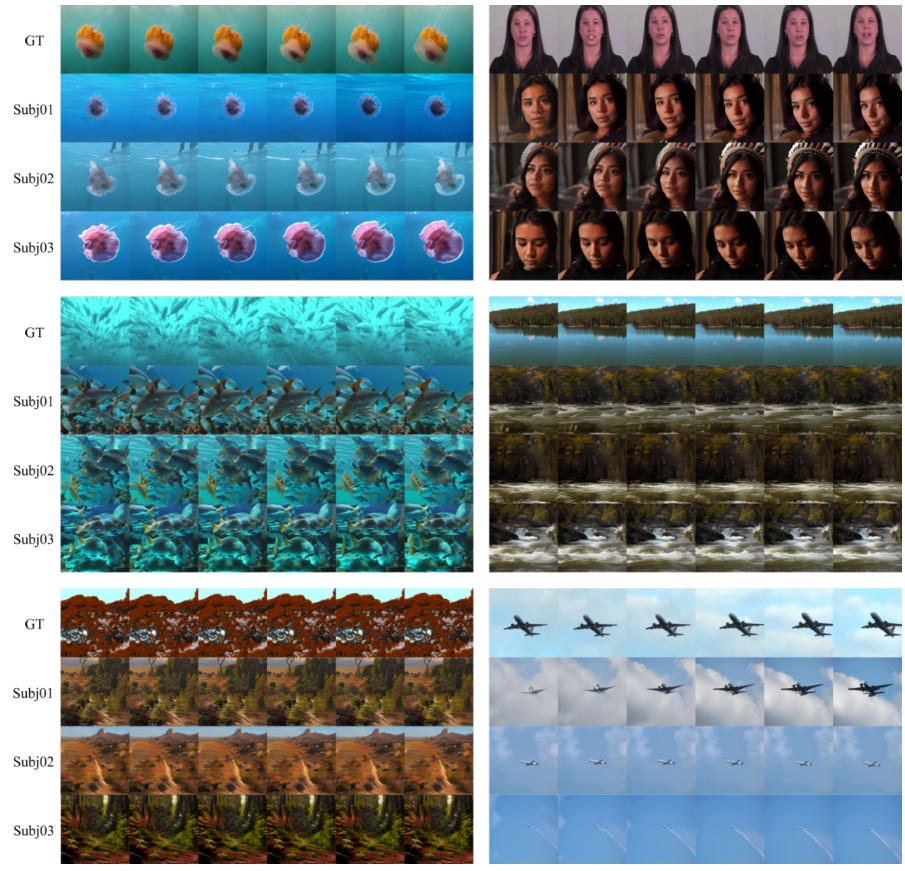

Figure 9: Visualization of successful reconstruction results by three subjects. We displayed 6 frames from each 16-frame video generated by fMRI. *Best viewed with zoom in*.

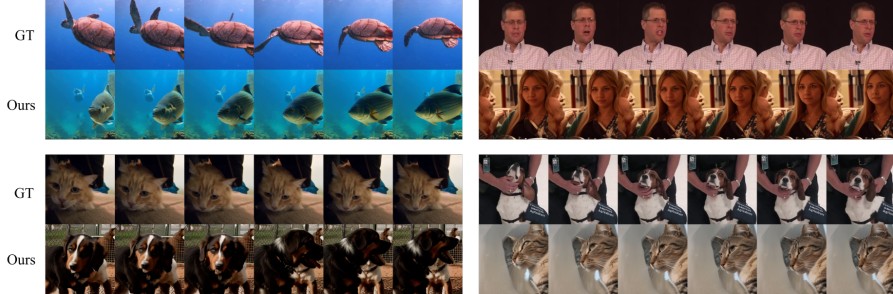

Figure 10: Visualization of some incorrect reconstruction results. We displayed 6 frames from each 16-frame video generated by fMRI. *Best viewed with zoom in*.

## C.4 More Ablation Results

The first frame of the blurry video provides structural information for the reconstruction of the keyframes and motion information for the generation of the video. Here, we perform the ablation visualization of the blurry video to observe its role in perception reconstruction. Although the building can still be generated with the blurry video removed as shown in Figure 11, the shape difference from the original image is too large, indicating the structural information provided by the blur video. In addition, with the addition of the blurred video guidance, the camera view is progressively upward, similar to the original image, but with the removal of the blurred video, the camera view is encircled, suggesting that the perception reconstruction provides motion guidance.

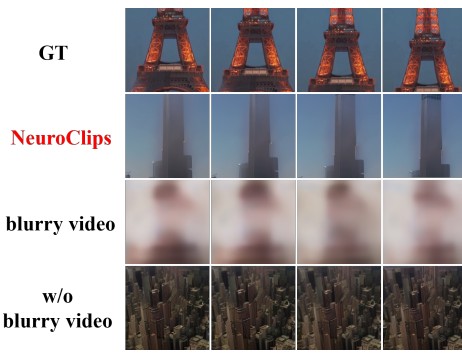

Figure 11: Visualization of ablation study.

## C.5 More Interpretation Results

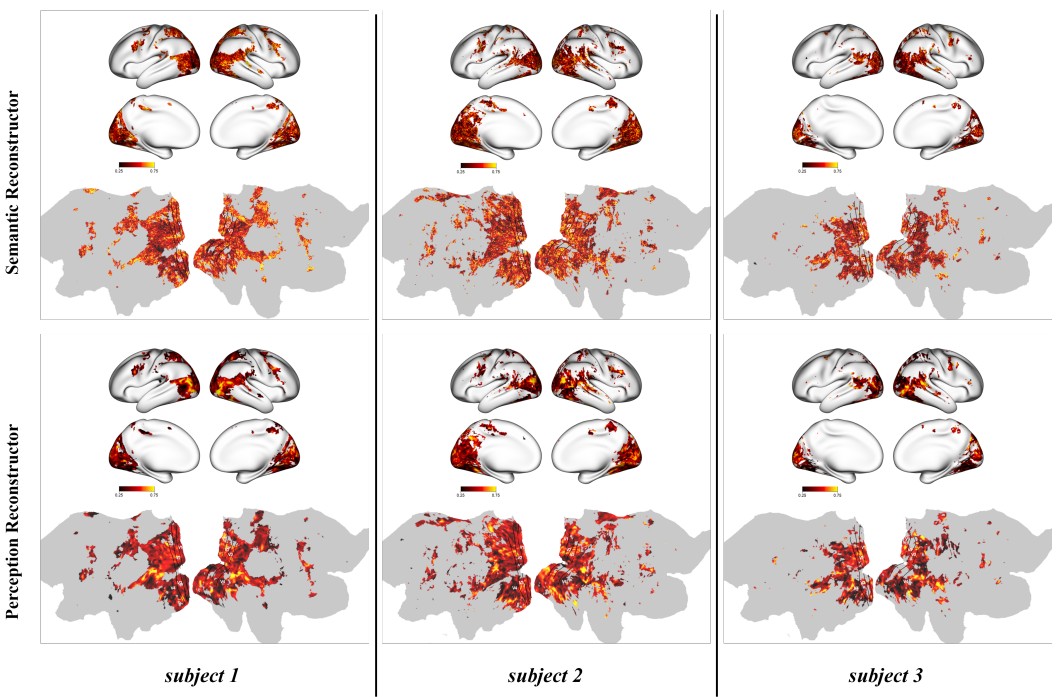

Figure 12: Additional visualization of voxel-wise weights. *Best viewed with zoom in.*

In Section 7, we visualized voxel-level weights on a brain flat map for subject 1. Here we provide detailed visualization results for all three subjects from the cc2017 dataset, as shown in Figure 12. As can be seen from the Figure, the three subjects learned similar weights in the Perception Reconstructor, which echoes the subjects' comparable SSIM metrics in Table 1 and demonstrates the existence of commonality in low-level vision in humans. However, the weights learned by subject 3 in the Semantic Reconstructor are different from the other subjects, which also leads to its image retrieval accuracy being at a lower value, as shown in Table 4. This may indicate that there were differences in the understanding of the video between subjects.

### C.6   Results of Retrieval Metrics

We further evaluate the performance of our Semantic Reconstructor using Top-1 keyframe retrieval accuracy and Top-1 fMRI retrieval accuracy. For keyframe retrieval, each test fMRI $\mathcal{Y}_c$ is first converted to an fMRI embedding $e_{\mathcal{Y}_c}$, and we compute the cosine similarity to its respective CLIP keyframe embedding $e_{\mathcal{X}_c}$ and 299 other randomly selected CLIP keyframe embedding in the test set. For each test sample, success is determined if the cosine similarity is greatest between the fMRI embedding and its respective CLIP keyframe embedding (aka top-1 retrieval performance, random chance is 1/300). The test set contains 1,200 fMRI-keyframe pairs, and we randomly divide it into 4 parts (thus each part contains 300 test pairs) for retrieval evaluation. We report the average retrieval accuracy among 4 parts. The same procedure is used for fMRI retrieval, except fMRI and keyframe are flipped. Table 4 displays the retrieval performance.

Table 4: The top-1 retrieval accuracy for each subject.

| Subject | Keyframe Retrieval ($\uparrow$) | fMRI Retrieval ($\uparrow$) |
|---|---|---|
| subject 1 | 23.1% | 20.7% |
| subject 2 | 26.3% | 19.8% |
| subject 3 | 17.3% | 15.9% |

Unlike the latest techniques for fMRI image reconstruction (where retrieval accuracy can reach more than 90%), *NeuroClips*' retrieval accuracy on the cc2017 dataset averages around 22%, which is at the lower end of the scale. Leaving aside the difference between fMRI in image stimuli and video stimuli, we found that the cc2017 test set had a large number of categories of objects that did not appear in the training set. This is likely to be the main reason for the low retrieval accuracy and is also the reason why most of the previous studies have focused on low-level visual reconstruction. In the future, a large-scale fMRI-video dataset more compatible with deep learning is worth looking forward to.

## D   Is *NeuroClips* Credible?

A big difference in *NeuroClips*' video generation inference is that we freeze the weights of the video diffusion model, while MinD-Video fine-tunes it. This raises the question of whether our generation relies exclusively on the knowledge base that the video diffusion model has been trained on, and is not consistent with the distribution of video training data related to fMRI.

Firstly, since our keyframes are control conditions as first frames, the content of our video generation is closely related to the keyframes, both semantically and structurally. So in *NeuroClips*, the video diffusion model will not rely entirely on its own semantic knowledge to generate videos, but on the keyframes. Secondly, for keyframe generation, keyframes are generated by SDXL unCLIP. The normal version of SDXL unCLIP generates images from a larger dimensional text representation space, which was fine-tuned for 6 epochs by Scotti et al. [2] on MSCOCO with image representations. The fine-tuned version of SDXL unCLIP provides a 256×1664 large dimensional CLIP representation space, which has a strong correspondence with the pixel space of the image, as shown in Figure 13. The SDXL unCLIP we use is this fine-tuned version. The videos captured from the cc2017 dataset used in this paper are from Videoblocks (https://www.videoblocks.com) and YouTube (https://www.youtube.com). These videos are not part of the COCO dataset, but SDXL unCLIP can still restore the original images as shown in Figure 13, demonstrating the equivalence of its representation space and pixel space. Since *NeuroClips* trained Semantic Reconstructor to align fMRI to this representation space, this also shows that our keyframe reconstruction is closely related to fMRI training.

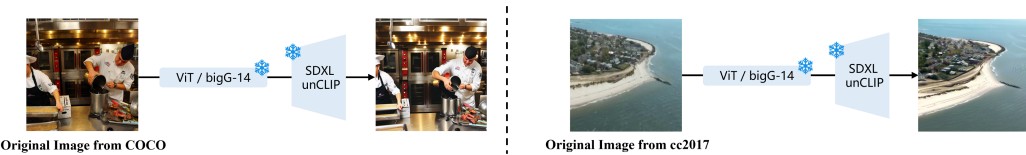

Figure 13: Visualisation of the generalization ability of SDXL unCLIP.

# E   Broader Impacts

Our research is dedicated to exploring the possibilities of neural decoding using deep learning techniques, which have a positive impact on the field of neuroscience. With the expansion of model scales and the improvement of corresponding hardware devices, this research will also make positive contributions to the field of brain-computer interfaces. However, this research also highlights the importance of personal privacy and security. Governments and research institutions should take appropriate measures to protect the privacy data collected and prevent any potential misuse. In our research, we use publicly available and de-identified datasets, so the study strictly adheres to relevant ethical requirements.

