# OpenReview forum: "NeuroClips: Towards High-fidelity and Smooth fMRI-to-Video Reconstruction"
_NeurIPS.cc/2024/Conference — NeurIPS 2024 oral_

### Official Review · Reviewer_oPVf · 2024-07-04

**Soundness:** 3
**Presentation:** 3
**Contribution:** 3
**Rating:** 7
**Confidence:** 4

**Summary:**

The paper introduces NeuroClips - a framework for reconstructing high-fidelity videos from fMRI data. It combines pixel-level and semantic-level visual learning through perception and semantic reconstruction pathways. The Perception Reconstructor (PR) ensures smoothness and consistency by creating a rough, continuous video, while the Semantics Reconstructor (SR) generates high-quality keyframes. These components are integrated into a video diffusion model, resulting in high-quality, detailed videos. Also, no additional post-diffusion training is required. NeuroClips sets a new standard in fMRI-to-video reconstruction, demonstrating significant advancements in semantic precision and pixel-level matching by improving SSIM by 128% and spatiotemporal metrics by 81%.

**Strengths:**

- Framework Design introduces a dual-component approach with a Semantics Reconstructor and a Perception Reconstructor to handle both high-level semantics and low-level perceptual details, respectively. Which is a novelty.

- Authors provide a comprehensive overview of existing methods for fMRI-to-video and fMRI-to-image tasks.

- The paper provides a thorough explanation of the NeuroClips framework, including the components, training procedures, and inference process. The implementation details are well-written so that readers can follow the technical aspects of the work. It also ensures reproducibility.

- The validation procedure is well-structured and clearly explained. The division of validation metrics in two (frames and video flow) allows one to better understand the performance of the NeuroClips framework.

- The newly introduced NeuroClips’ SR which allows the generation of longer videos. The novelty of the Neuroclip pipeline, the high validation metrics scores and creative approach to dealing with fMRI data makes the proposed framework significant.

**Weaknesses:**

- The multi-fMRI fusion strategy is briefly described, but the implementation details and the rationale behind specific design choices are not fully elaborated. I suppose the scheme of the multi-fMRO strategy in the supplement would increase the clarity of this paragraph.

-  As it can be seen from the examples provided with the code repository, the proposed framework does not account for a change of the scene in the video (which was briefly mentioned by the authors in the Limitations). The pipeline with chosen keyframe might hinder the NeuroClips  to catch this rapid change in the video. No ablation is done in this direction, which could explain how neural NeuroClips decodes fMRI signals.

- The paper primarily evaluates the method on a specific dataset (with only 3 patients), which, obviously, may not fully capture the diversity of real-world video content and fMRI recordings.This should be at least mentioned in the Discussion (or Conclusion) of the work.

 - The neurobiological justification of keyframe usage seems ambiguous. The improvement of text clearance and up-to-date references would increase the significance of the work.

**Questions:**

- The paper mentions accounting for the hemodynamic delay (BOLD signal delay of approximately 4 seconds), there is limited discussion on how this delay specifically impacts the reconstruction quality and temporal alignment with video frames.

- The use of ridge regression to map fMRI signals to lower-dimensional embeddings assumes a relatively linear relationship between neural activity and visual stimuli. However, the brain's processing of visual information is highly nonlinear and complex. At least, it should be mentioned in the Limitations too.

- Fig 12: The weights of subject 3 are really similar for PR and SR tasks, can you elaborate on it?
 - Fig 2: Example images are too small to perceive it even with a zoom, I recommend to make them bigger (as in Fig 7).
- I recommend adding the major limitations in the Conclusion section of the main text.

- Authors should include in the Supplement section which software was used to build brain maps with weights and if any data pre-processing/normalization was applied.

- Lines 75-76: “NeuroClips achieves a 128% improvement in SSIM and an 81% improvement in spatiotemporal metrics and also performs better overall on most video semantic-level metrics.” -  report not only the percent of improvement, but also the values of the metrics.

- In the frame validation the main image quality metrics are PSNR and SSIM. Authors should mention that those metrics have flaws (Zhang et al, The Unreasonable Effectiveness of Deep Features as a Perceptual Metric, 2018). Will it be possible to consider other quality evaluation metrics? Like, for example, Visual information fidelity metrics (Sheikh et al., Image information and visual quality, 2006). It is also interesting why PSNR and SSIM were used for frames-based evaluation but not for video-based evaluation with modification to catch spatio-temporal data (ST-PSNR and ST-SSIM)? Another question is why authors choose to evaluate performance only with “CLIP image embeddings on each frame” and omit other metrics like Fréchet Video Distance (FVD), MSE or already mentioned ST-PSNR and ST-SSIM?

- Fig 12: The visualization of voxel weights on a brain flat map to interpret neural activity is a significant strength. However, now it is difficult to comprehend the differences in weights from the image with naked eyes. The addition of a third row with the difference between the weights of participants can increase the clarity of what authors are trying to say.

- The study focuses primarily on the visual cortex. While this is appropriate for the simplest visual stimuli (i.e., dots, simple geometrics shapes, etc.), it may limit the generalization of the approach to other types of brain activity or cognitive functions. Discussing potential extensions to other brain regions and types of neural data could provide a more comprehensive neurobiological perspective.

-  Lines 316-318: “From the quantitative results, it can be seen that there is a trade-off between semantic and perception reconstruction” - there is no clear explanation or reasoning why there is a trade-off between SR and PR

- Line 38: “However, the visual decoding of MinD-Video significantly diverges from the brain’s visual system, exhibiting limitations in perceiving continuous low-level visual details.” - the more precise listing of what hinders MinD-Video to perceiving continuous low-level visual details would be more constructive, rather than comparison with the brain's visual system.

- Lines 41-42: “Notably, the human brain perceives videos discretely [8, 9] due to the persistence of vision [10, 11] and delayed memory [12].”
	1. discretely -> discreetly
	2.  Was it intentional to use the paper [10] of 1892 year for the “persistence of vision” concept? Also, the fact that the brain has a “persistence of vision” was criticized. Actually, the second citation you’ve used [11] is the critique of this concept (J Anderson et al., The myth of persistence of vision revisited, 1993). The citations of the novel research papers are required to justify this statement.

The paper will be significantly improved by more valid justification of claimed neurobiological concepts.


Minor:
- Line 644: “However, these limitations will not be tackled overnight” - this phrase is informal and somewhat colloquial, which may not be entirely appropriate for a scientific paper
- Line 56: “This process is reflected in the fMRI signal” - can you provide the link to the paper supporting this statement?
- Line 129: “...loss to train the PR, the overall loss L_{PR} of PR can be donated as…” - “donated” is a confusing word to use, please use another word.

-  Lines: 222-223: “Since the technical field of long video generation is still immature, we chose a more straightforward fusion strategy that” - can not see why immatureness of the field can be used as a reason to use a straightforward strategy. Please, make the reason for usage more understandable.

- Figure 4 in page 8: I recommend not to use comic sans font in images.

- Lines 596-597: “It proves that there may be a large difference in the understanding of the video” - please refrain from using the word “prove” here, since it implies comprehensive research or, at least, references to other works that proves it.

- Could you report the number of classes you have at least in training and testing dataset (for the “N-way top-K accuracy classification test as the semantics-level metric”)

- Line 55-56: “generating high-level images in the cerebral cortex” - it is also knows that brain interpolates the seen scenes (Vacher, Jonathan, et al. Texture interpolation for probing visual perception, 2020), which could be used as justification to use keyframe approach.

**Limitations:**

Important limitation - the dataset of only 3 participants - was not mentioned. This may result in varying interpretations with a careful review by a neurobiologist. Additional evaluation is needed with other datasets and more participants.

---

> ### Author Rebuttal · Authors · 2024-08-04
>
> We sincerely appreciate your careful review. In particular, you have provided us with more than twenty constructive suggestions and insightful questions, which is extremely precious at a time when the quality of reviews in the community is deteriorating. We value each of your suggestions and provide the following responses:
>
> > **Weakness 1: Details of Multi-fMRI Fusion**
>
> We acknowledge and accept your suggestion. Considering that Multi-fMRI Fusion is merely an extra merit of NeuroClips and the page limitations, we were unable to present more details regarding multi-fMRI fusion.
> Thanks to the design of keyframes in the Semantic Reconstructor (SR), Multi-fMRI Fusion can be effectively achieved. In this study, we obtain the CLIP representations of reconstructed neighboring keyframes and train a shallow MLP based on the representations to distinguish whether two frames share the same class. The exact process is also shown as the GIF in the code repository. This training process operates at the image level. Despite our efforts to assess whether neighboring fMRI frames belong to the same scene class at the fMRI frame level using an MLP with fMRI frames as inputs, aiming for fMRI anomaly detection, our results were suboptimal, as illustrated in the following table. As can be seen, the fMRI level analysis frequently led to excessive false fusions. Conversely, established techniques for categorizing images perform reliably, particularly within the proficient CLIP space we employ.
> We promise to include more technical details in the final version. Thank you for your suggestion.
>
> | Fusion-level            | Subject 1 | Subject 2 | Subject 3 |
> | ----------------------- | --------- | --------- | --------- |
> | fMRIs                   | 59.7%     | 58.8%     | 57.2%     |
> | Reconstructed keyframes | **86.3%** | **87.1%** | **85.6%** |
>
> > **Weakness 2: Rapid Scene Change**
>
> **We will present four perspectives on the inability of NeuroClips to perceive rapid changes in the scene**
>
> * **Regarding Dataset.** The rapid scene change depicted in the code repository is a **distinct** alteration influenced by **human intervention**, diverging from authentic real-world visuals. In the CC2017 dataset, video sequences were randomly divided into discrete segments, then amalgamated and presented to subjects. Such spliced videos are seldom encountered in everyday visual encounters, as scenes typically unfold continuously. This is what we have mentioned in Limitation about *cross-scene fMRI*.
>
> * **From the fMRI Data.** We must acknowledge that **continuous** scene transitions occur in actual visual perception, such as when a person turns their head. However, it remains unknown whether and how rapid scene changes occurring within an fMRI frame (e.g., 2 seconds) are reflected in the fMRI signal due to its intricate nature. Consequently, decoding scene changes from fMRI data poses a significant challenge.
>
> * **From Keyframe Design.** When scene changes occur within one fMRI frame (e.g., 2s), the design of the keyframes may adequately capture it, although the blurred video would be continuous. However, the keyframe-based method to guide the generation of continuous video can serve as a foundation for the subsequent design of decoding two consecutive scenes separated rapid scene change.
>
> * **From Cross-scene Diffusion Models.**  Presently, most of the current video generation diffusion models are not capable of very silky smooth scene switching, like SORA, to the best of our knowledge. We believe that, from a technical point of view, the issue can be better alleviated if consecutive semantics can be decoded from a single fMRI frame, or if the temporal resolution of fMRI can be refined.
>
> Thanks again for your suggestions. We will provide the results of some of our exploratory experiments and a detailed discussion in the final version.
>
> > **Weakness 3: Generalization Capability**
>
> **Diverse real-world video content**. The videos in the CC2017 dataset used are, somehow, sufficiently reflective of real-world visual experiences. As the original dataset paper states 'All video clips were chosen from Videoblocks and YouTube to be diverse yet. For example, individual video clips showed people in action, moving animals, nature scenes, outdoor or indoor scenes, etc' [1].
>
> **Diversity of fMRI recordings.** To ensure consistency with the baselines [2] and to make a fair comparison, we experimented on the dataset, which unfortunately has only 3 subjects. We appreciate you pointing out that this should at least be mentioned in Limitations, and we will add it in the final version. Publicly available datasets for fMRI-to-video reconstruction are valuable and not easy to find. Until now, we discovered that the Algonauts 2021 [3] dataset can also be used for video reconstruction, but it is a great pity that this dataset is currently in an unpublished stage. To show the generalization capability of our approach, we finally chose to perform fMRI-to-image reconstruction on the Natural Scenes Dataset (NSD) [4] to assess the keyframe effect of our Semantic Reconstructor (SR). The visual results of the reconstructed image and groundtruth images are shown in the **`PDF`** appendix. Notably, our method exhibited satisfactory reconstruction outcomes even when applied to fMRI data with a distinct distribution, signifying the generalization capabilities of NeuroClips.
>
> **Reference**
>
> [1] Neural encoding and decoding with deep learning for dynamic natural vision, *Cerebral cortex 2018*
>
> [2] Cinematic mindscapes: High-quality video reconstruction from brain activity, *NeurIPS 2023*
>
> [3] The algonauts project 2021 challenge: How the human brain makes sense of a world in motion, *2021*
>
> [4] A massive 7T fMRI dataset to bridge cognitive neuroscience and artificial intelligence, *Nature neuroscience 2022*

---

> ### Author Response · Authors · 2024-08-04
> **Response to Weaknesses of Reviewer oPVf**
>
> > **Weakness 4: Neurobiological Justification**
>
> Thank you for sharing your expert advice from neuroscience.
>
> In numerous studies within **cognitive neuroscience** and **computational neuroscience**, researchers have delved into the mechanisms related to the **brain's visual information processing** and memory functions. Building on these studies, we innovatively propose using keyframes to guide our research, addressing the issue of frame rate mismatch between visual stimuli and fMRI signals, and enhancing the model's accuracy in fMRI-to-Video Reconstruction.
>
> Specifically, [1] demonstrate that **'key-frames'** play a crucial role in how the human brain recalls and connects relevant memories with unfolding events. In [2], a novel video abstraction paradigm which use the brain response reflected by fMRI to guide the extraction of visually informative segments from videos was proposed to quantitatively reveal the attentional engagement of human brain in the comprehension of video. In [3], the key frames in a video clip were used to extract these features, with the combined features from all **keyframes** representing the entire **video clip**. [4] provided a framework and explanation for video summarization.
>
> We will revise and update our text clearance and references in the final version.
>
> **Reference**
>
> [1] Brain mechanisms underlying cue-based memorizing during free viewing of movie Memento, *NeuroImage 2018*
>
> [2] Video abstraction based on fMRI-driven visual attention model, *Information sciences 2014*
>
> [3] Bridging the semantic gap via functional brain imaging, *IEEE Transactions on Multimedia 2011*
>
> [4] A comprehensive survey and mathematical insights towards video summarization, *Journal of Visual Communication and Image Representation 2022*

---

> ### Author Response · Authors · 2024-08-07
> **Response to Questions of Reviewer oPVf**
>
> > **Question 1: Hemodynamic Delay**
>
> Indeed, a considerable number of **exploratory experiments** have been conducted on the topic of **hemodynamic delay**, which has also been considered in Mind-Video. In the current version of NeuroClips, which employs a fixed 4-second delay, it was observed that the semantics of some of the generated keyframes exhibited latency, particularly in instances where the video clips of the scene were of a **greater duration**. Accordingly, a **sliding window** comprising two or three fMRI frames was devised to actively learn the aforementioned delay. However, it was discovered that employing a sliding window resulted in a **notable reduction** in the final evaluation metrics, with a **more pronounced negative impact**, particularly in the case of shorter video clips. It may be the case that longer videos have a more enduring effect on human brain fMRI signals. In light of the experimental outcomes, we ultimately opted to discard this methodology and instead fix the delay.
>
> > **Question 2: Ridge Regression**
>
> As you mentioned, the human brain processes information in a highly complex and non-linear way. However, empirical evidence [1, 2, 3] underscores the **effectiveness and sufficiency of linear mapping** for achieving desirable reconstruction outcomes. Notably, **complex nonlinear models will easily overfit to fMRI noise**, leading to poor performance in the test set [4]. We will add more discussion in the Method Section.
>
> **Reference**
>
> [1] Reconstructing the mind's eye: fMRI-to-image with contrastive learning and diffusion priors, *NeurIPS 2023*
>
> [2] High-resolution image reconstruction with latent diffusion models from human brain activity, *CVPR 2023*
>
> [3] MindEye2: Shared-Subject Models Enable fMRI-To-Image With 1 Hour of Data, *ICML 2024*
>
> [4] Through their eyes: multi-subject Brain Decoding with simple alignment techniques, *Imaging Neuroscience 2024*
>
> > **Question 3: Voxel Weight Visualization**
>
> In Figure 12, each column represents the PR and SR visualization weights for the same subject. The voxel distribution is the same in each column since the **voxels initially selected by each subject are fixed**. For each subgraph, the weights of the voxels were visualized after normalization, with **brighter regions** representing **higher weights** and **darker regions** representing **lower weights**. For subject 3, the distribution of voxel weights in PR and SR is quite different. For example, in the right region of V4 of the right brain, PR is brighter and SR is darker. We appreciate your suggestion to add another line to visualize and highlight the **difference** between PR and SR. In this way, we can better show the difference in voxel weights between the two modules, and we will add it to the final Supplement.
>
> > **Question 4: Larger Example Images**
>
> We will modify it to a vector figure and adjust the number of images to make it clearer.
>
> > **Question 5: Major Limitations**
>
> Since NeurIPS allows an additional page to be added to the final version, we will move Limitation from the appendix to the main body. Thanks for your suggestion.
>
> > **Question 6: Visualization Software**
>
> We use **Connectome Workbench** from Human Connectome Project (*HCP*),  and flatmap templates were selected as 'Q1-Q6_R440.L.flat.32k_fs_LR.surf.gii' and Q1-Q6_R440.R.flat.32k_fs_LR.surf.gii' . Additionally, the cortical parcellation was manually delineated by **neuroimaging specialists** and **neurologists**, and aligned with the public templates in **FreeSurfer software** with verification. We **normalised** the voxel weights, scaling them to between 0 and 1. Finally, to show a better comparison, the **Colorbar** was chosen to be 0.25-0.75. We will add this note to the final Supplement section.
>
> > **Question 7: Metric Improvement**
>
> We'll add extral analysis to indicate the improvement values on the metrics.

---

> ### Author Response · Authors · 2024-08-07
> **Response to Questions of Reviewer oPVf**
>
> > **Question 8: More Metrics**
>
> The two new metrics ST-SSIM and ST-PSNR you mentioned are very interesting and enlightening, and we have carried out comparative experiments with MinD-Video in terms of the two metrics. We also evaluated NeuroClips with Visual Information Fidelity (VIF) metrics, and the results are shown in the table below (All results are averages of 3 subjects). Notably, NeuroClips' performance at the ST-level far exceeds that of MinD-Video, and even more than pixel-level, which further indicates that NeuroClips has a smoother video reconstruction capability.
>
> | Method|ST-SSIM|SSIM|ST-PSNR|PSNR|VIF|
> | - | - | - | - | - | - |
> | MinD-Video |0.489|0.171|11.595|8.662|0.113|
> | NeuroClips | **0.785** | **0.390** | **17.200** | **9.211** | **0.170** |
>
>
> As commonly used evaluation metrics in the field of **video generation**, *Fréchet Video Distance (FVD)* is more often used to assess the performance of video diffusion generation models. However, since we **freeze** the pre-trained parameters of the advanced generation model *Animatediff* [1], the effect of our video generation model must be better than the previous video models. Therefore, the evaluations on these video metrics may not be a **fair** one. Instead, we consider the CLIP rerepsentations for evaluation. Note that the consistency in CLIP representation space is more revealing of the degree of **semantic consistency**, reflecting the superiority of Semantic Reconstructor (SR) in NeuroClips.
> In the table above, we conducted an evaluation using the Visual Information Fidelity metric (VIF), instead of the FVD metric. When assessing FVD, all of Pytorch's open-source methods necessitate a video frame rate exceeding 10fps due to their need for some level of downsampling. Despite NeuroClips making significant advancements in frame rate, it falls to meet this requirement.
>
> [1] Animatediff: Animate your personalized text-to-image diffusion models without specific tuning, *ICLR 2024*
>
> > **Question 9: Difference Voxel Weight Visualization**
>
> We will add a third row with the difference between the weights of subjects as described in *Question 3*, thanks again!
>
> > **Question 10: Voxel Selcetion**
>
> In fact, we did **not** select voxels **specifically for the visual cortex**. We calculated the voxel-wise correlation between the fMRI voxel signals of each training movie repetition for each subject. **The significant voxels**(Bonferroni correction, **P < 0.05**) were considered to be **stimulus-activated voxels** and used for subsequent analysis, as described in the pre-processing paragraphs of **Section 4.1**. We agree with you that other brain regions may also contribute to video decoding as well, so we expanded the range of significance (**MinD-Video using P < 0.01**), and the voxels used in NeuroClips are more than MinD-Video. The following table shows the number of voxels selected by the two methods.
>
> |Method|Subject 1|Subject 2| Subject 3|
> |-|-|-|-|
> | MinD-Video|6016| 6224|3744|
> | NeuroClips|13447|14828|9114|
>
> We believe that our voxel-selective paradigm is more easily migrated to other fMRI decoding tasks and provides a more comprehensive neurobiological perspective.
>
> > **Question 11: Trade-off Between PR & SR**
>
> Due to the parallel design of our Semantic Reconstructor (SR) and Perception Reconstructor (PR), SR focuses more on the decoding of **video semantics**, while PR is more geared towards the reconstruction of **pixel-level information**. SR can significantly improve semantic-related metrics, and PR can improve pixel-level related metrics. So in the end, when the two are combined together, there is a trade-off between semantic and perception reconstruction. During training, the Video Diffusion model will achieve a compromise effect between semantic and perception reconstruction. We will provide more discussions and deeper insighs on the compromise effect.
>
> > **Question 12: Low-level Visual Details**
>
> Thanks for your suggestion. We willl modify it to '***MinD-Video lacks design of low-level visual detailing, so it significantly diverges from the brain’s visual system, exhibiting limitations in perceiving continuous low-level visual details.***'
>
> > **Question 13: Persistence of Vision**
>
> Thanks for your suggestion. Regarding the persistence of vision, we will careful check the references and discussions and supplement more supporting references outlined below.
>
> **Reference**
>
> [1] Ultra-High Temporal Resolution Visual Reconstruction From a Fovea-Like Spike Camera via Spiking Neuron Model, *TPAMI 2023*
>
> [2] CaRiNG: Learning Temporal Causal Representation under Non-Invertible Generation Process, *Arxiv 2024*
>
> [3] Persistence Of Vision Display-A Review, *IOSR-JEEE e-ISSN 2015*
>
> [3] POV: Persistence of Vision: International Journal of Ethics in Engineering & Management Education
>
> [4] Persistence of vision: the interplay of vision, *Vision, Memory and Media 2010*

---

> ### Author Response · Authors · 2024-08-07
> **Response to Minors of Reviewer oPVf**
>
> > **Minor 1**
>
> We would modify this sentence to "***The alleviation of these limitations will require joint advances in multiple areas and significant further effort will be required.***" Thank you.
>
> > **Minor 2**
>
> fMRI measures brain activity by detecting changes in blood flow, and can thus reflect and quantify brain activity evoked by visual stimuli which has been widely applied in studies. The following are relevant references:
>
> **Reference**
>
> [1] Reconstructing perceived images from human brain activities with Bayesian deep multiview learning, *TNNLS 2019*
>
> [2] Survey of encoding and decoding of visual stimulus via FMRI: an image analysis perspective, *Brain imaging and behavior 2023*
>
> [3] Compressive spatial summation in human visual cortex, *Journal of Neurophysiology 2013*
>
> [4] fMRI evidence for areas that process surface gloss in the human visual cortex, *Vision research 2015*
>
> [5] A comparison of fMRI adaptation and multivariate pattern classification analysis in visual cortex, *Neuroimage 2010*
>
> [6] Spontaneous activity associated with primary visual cortex: a resting-state FMRI study, *Cerebral cortex 2008*
>
> [7] The human visual cortex, *Annual review of neuroscience 2004*
>
> > **Minor 3**
>
> Thanks for your advice!  We will replace ‘donated’ with **‘described’** in the final version.
>
> > **Minor 4**
>
> In the context of the current **diffusion model** and **attention-based transformer model** as the dominant models for image generation, the computational overhead required for image generation models is sufficiently large. The content of video grows linearly with the number of frames, so the technical field of long video generation is still immature.
>
> We value your opinion and also believe that a clearer explanation is needed here. Therefore, we will revise the above statement to make it more understandable.
>
> > **Minor 5**
>
> Thank you for your professional suggestion. We decide to change the font in the Figure 4 to 'Times New Roman' in the final version.
>
> > **Minor 6**
>
> Thanks for your kind reminder. Upon careful consideration, we acknowledge that using **'prove'** is indeed inappropriate since the paragraph is an exposition of the results of the experiment. We agree with your suggestion and will change the sentence to ***"This may indicate that there were differences in the understanding of the video between subjects."***
>
> > **Minor 7**
>
> The classifiers we use are **frozen pre-trained classifiers**, so the total number of categories is fixed independent of the CC2017 dataset. The image classifier is an ImageNet classifier, pre-trained on ImageNet-1K[1] hence **1000 image classes**. The video classifier based on VideoMAE [2] is trained on Kinetics-400 [3], an annotated video dataset with **400 classes**, including motions, human interactions, etc.
>
> **Reference**
>
> [1] Imagenet: A large-scale hierarchical image database, *CVPR 2009*
>
> [2] Videomae: Masked autoencoders are data-efficient learners for self-supervised video pre-training, *NeurIPS 2022*
>
> [3] The kinetics human action video dataset, *ArXiv 2017*
>
> > **Minor 8**
>
> Thank you for your valuable addition. We will carefully review this literature and include it as a justification for using the keyframe approach in the final version.
>
> &nbsp;
> ***
>
> We would like to extend our heartfelt gratitude to you! Our sincerest thanks! Your insights and suggestions from a neuroscience perspective have provided us with numerous inspiring ideas, greatly benefiting not only our current work but also our future research endeavors. Additionally, your detailed feedback has made our submission more solid and complete. We hope our response adequately addresses your questions, and we eagerly look forward to further communication and discussions.
>
> Once again, we sincerely express our deepest gratitude and highest respect for your effort and time!
>
> Best wishes,
>
> All authors of Submission 351

---

> > ### Comment · Reviewer_oPVf · 2024-08-09
> > **Post-rebuttal thoughts**
> >
> > Thank you for a rebuttal as thoroughly prepared as the submission itself. Despite the limitation of a low-n study and some debatable conclusions for the neurobiology (which had motivated my original score) AND given the authors keep their 5 rebuttal promises for the camera ready, I am now willing to give an extra point and be a proponent of this work during the discussion with ACs. Thank you for interesting read and good luck!

---

> > > ### Author Response · Authors · 2024-08-10
> > > **Heartfelt Thanks**
> > >
> > > We greatly appreciate your constructive feedback and meticulous review. And it is good to know that our response has addressed some of your concerns. Although some others have not yet reached a mutual agreement, we are committed to resolving them promptly to ensure the high quality of the work. Lastly, we would like to express our sincere gratitude for your increased rating and further support towards our work! We hope this paper achieves satisfactory results, not in vain of your efforts and suggestions.
> > >
> > > Best wishes,
> > >
> > > All authors of Submission 351

---

### Official Review · Reviewer_mcBd · 2024-07-11

**Soundness:** 3
**Presentation:** 4
**Contribution:** 3
**Rating:** 8
**Confidence:** 4

**Summary:**

The proposed framework NeuroClips introduces a strong pipeline for fMRI-to-Video reconstruction in the field of Brain Visual Decoding. The Perception Reconstructor(PR) maintains the motion of the video and the Semantic Reconstructor(SR) ensures the semantic information of the video. Multi-fMRI Fusion raises upper video length limit, and overall model achieves impressive results.

**Strengths:**

1. The paper is clearly written and easy to read, with clear diagrams and charts.
2. Experiments and discussions are conducted extensively, including rich video reconstruction and neural interpretation visualization content to validate the model's performance, which strengthens the results and the paper in general.
3. The proposed rough-video reconstruction in Perception Reconstructor(PR) and the strategy of Multi-fMRI Fusion are generally innovative, and have greatly contributed to break through the original low frame rate and fixed length of 2s video limitations in previous methods. In addition, the designs of NeuroClips for textual semantics and pre-trained diffusion models for video generation are also unique. Considering NeuroClips' powerful results and methods, I think it can be a strong baseline for the emerging fMRI-to-video reconstruction field.

**Weaknesses:**

1. I browsed the anonymous site, and the generated results are impressive. However, the lighting of some of the reconstructed videos varies considerably compared to the ground truth, which can also be seen on the right side of Figure 2, and the authors need to rationalize this.
2. Existing state-of-the-art video generation models can generate high-frame rate videos such as 24 fps for up to 1 min or even longer, however NeuroClips at this stage has not yet reached this level.

**Questions:**

1. As discussed in mind-video [1], the nature of hemodynamic response has been considered and specific modules are designed, which seems not to be included in NeuroClips. Are there other considerations or is there no need to take into account for the BOLD signals?
2. Since a number of cross-subject models already exist in the image reconstruction field [2], does NeuroClips need to train a separate model for each subject?
3. As you mention in the limitation section of the appendix, the cc2017 dataset test set contains too many no-show categories, and I'm curious if the unsatisfactory results of previous methods are more due to the low quality of the dataset, less than the method itself?
4. Why text contrastive loss is placed after diffusion prior and not before like in [3]?

[1] Chen, Zijiao, Jiaxin Qing, and Juan Helen Zhou. "Cinematic mindscapes: High-quality video reconstruction from brain activity." Advances in Neural Information Processing Systems 36 (2024)

[2] Scotti, Paul S., et al. "MindEye2: Shared-Subject Models Enable fMRI-To-Image With 1 Hour of Data." arXiv preprint arXiv:2403.11207 (2024).

[3] Sun, Jingyuan, et al. "NeuroCine: Decoding Vivid Video Sequences from Human Brain Activties." arXiv preprint arXiv:2402.01590 (2024).

**Limitations:**

Yes.

---

> ### Author Rebuttal · Authors · 2024-08-06
>
> We would like to thank you for taking the time to review our work. We value each of your suggestions and provide the following responses:
>
> > **Weakness 1: Light Variations**
>
> We highlight that **light variations** are also a feature that distinguishes video from images. Our initial hypothesis was that this phenomenon was caused by the presence of partial light variations in the blurred video of the Perception Reconstructor (PR). However, subsequent experiments involving the **removal** of the blurred video revealed that the effect persisted. This may be caused by the pre-training of AnimateDiff, where a corresponding discussion will be further explored in the final version of the paper. It is still important to emphasise that although the phenomenon exists, it is only observed in a **limited number** of videos.
>
>
> > **Weakness 2: Longer and High-frame Videos**
>
> + **From the Dataset.** After counting the video clips in our dataset, we discovered that the longest video spanned merely around 10 seconds. Hence, there is no need to reconstruct videos that extend up to a minute in duration.
>
> + **From an availability standpoint.** Currently, the most cutting-edge video generation technology has now advanced to create longer videos, like **`Sora`**. However, the quality of the generated content still raises concerns due to the current limitations of the technology. Additionally, it's important to highlight that the majority of these technologies are **not open source.**
>
> + **From a Research Perspective.** fMRI-to-video reconstruction is an emerging field. At this critical stage, we believe that the applicability and scalability of innovative methods are of greater consequence. Empirically, once longer blurred videos or video generation tools are available, **NeuroClips** can generate **24-frame and longer videos**. However, this will **remarkably increase the GPU usage**. From a research standpoint, we believe that the current reconstructed video is impressive and sufficient.
>
> > **Question 1: Hemodynamic Delay**
>
> Indeed, a considerable number of **exploratory experiments** have been conducted to study **hemodynamic delay**, a topic also considered in Mind-Video. In the current version of NeuroClips, with a fixed 4-second delay, it was observed that the semantics of some of the generated keyframes exhibited latency, particularly in instances where the video clips of the scene were of a **greater duration**. To address this, a **sliding window** comprising two or three fMRI frames was devised for the purpose of actively learning the aforementioned delay. It was discovered that the application of a sliding window resulted in a **notable reduction** in the final evaluation metrics, with a **more pronounced negative impact**, particularly in the case of shorter video clips. It may be the case that longer videos have a more enduring effect on human brain fMRI signals. In light of the experimental outcomes, we ultimately opted to discard this approach and instead maintain a fixed delay.
>
> > **Question 2: Cross-Subject**
>
> Yes, it is necessary to train a **distinct** model for each subject when using NeuroClips. As cross-subject approaches in this area still fail to achieve satisfactory results, we finally choose to utilise and explore a **single-subject** model in this paper. However, in light of the recent advancements in **fMRI-to-image reconstruction**, a series of cross-subject models have emerged [1, 2, 3]. We believe that extending NeuroClips to encompass cross-subject content will be a **promising avenue** for future research.
>
>
> **Reference**
>
> [1] Mindbridge: A cross-subject brain decoding framework, *CVPR 2024*.
>
> [2] Psychometry: An omnifit model for image reconstruction from human brain activity, *CVPR 2024*.
>
> [3] MindEye2: Shared-Subject Models Enable fMRI-To-Image With 1 Hour of Data, *ICML 2024*
>
> > **Question 3: Low Quality of Dataset**
>
> Since both NeuroClips and the previous baselines are founded upon the **same dataset**, the ensuing comparison is deemed to be **equitable**. The assertion that the CC2017 dataset is of poor quality is intended to **highlight** the difficulty in achieving impressive fMRI-to-video reconstruction results in **comparison to fMRI-to-image reconstruction**. The capacity to produce superior outcomes on datasets of inferior quality serves to illustrate the **resilience** of the NeuroClips' method. It would be of interest to ascertain whether NeuroClips could achieve even more impressive results with a higher-quality dataset.
>
>
> > **Question 4: Enhancement from Text Modality**
>
> Considering that the diffusion prior loss is rooted in MSE at the representation level, this prior is inherently **unstable** and the semantic information is susceptible to **bias**. It is important to recognise that text modality possesses its own **distinctive characteristics** and **robust semantic support**, which serve to complement the image representation space. To enrich the semantic depth of the representation, we strategically place text assistance subsequent to the prior.
>
> &nbsp;
> ***
> You have offered many constructive and valuable suggestions, making our submission more solid and complete. Once again, we sincerely express our best gratitude for your effort and time!
>
> Best wishes,
>
> All authors of Submission 351

---

> > ### Comment · Reviewer_mcBd · 2024-08-12
> >
> > I would like to thank the authors for their responses. It is good to see that my concerns have been sufficiently addressed. I think the submission is solid, and I will raise my rating.

---

> > > ### Author Response · Authors · 2024-08-12
> > >
> > > We greatly appreciate for your constructive feedback. We would like to express our sincere gratitude for your increased rating and further support towards our work! We hope that this paper achieves satisfactory results, not in vain of your efforts and suggestions.
> > >
> > > Once again, thank you, Reviewer mcBd.
> > >
> > > Best wishes,
> > >
> > > All authors of Submission 351

---

### Official Review · Reviewer_85K8 · 2024-07-12

**Soundness:** 3
**Presentation:** 3
**Contribution:** 3
**Rating:** 7
**Confidence:** 4

**Summary:**

This paper proposes NeuroClips, a framework that decodes high-fidelity and smooth video from fMRI. NeuroClips uses a semantics reconstructor for video keyframes to ensure semantic accuracy and consistency, and a perception reconstructor for capturing low-level perceptual details, ensuring video smoothness.

**Strengths:**

NeuroClips is the framework to decouple high-level semantics and low-level perception flows for fMRI-to-video reconstruction, achieving high-fidelity and smooth video outputs.
The framework addresses the temporal resolution gap between fMRI and video data, ensuring smooth and consistent video outputs through innovative modules like Inception Extension and Temporal Upsampling.

**Weaknesses:**

The video captures the semantic meaning well but fails to accurately follow the ground truth movement.

**Questions:**

- Why is there no model or loss function for movement(motion)?
- Are you willing to make the code publicly available?

**Limitations:**

As mentioned above, motion reconstruction has not been resolved.

---

> ### Author Rebuttal · Authors · 2024-08-05
>
> We would like to thank you for taking the time to review our work. Your effort has ensured that our submission received adequate attention and review. We address the questions and clarify the issues accordingly as described below.
>
> > **Weakness 1: NeuroClips Fails to Accurately Follow the Ground Truth Movement**
>
> + Thanks for pointing out your concern. The fMRI-to-video decoding task presents significant challenges related to both high-level semantic comprehension and low-level motion perception. On the one hand, fMRI data is characterized by its high dimensionality, with hundreds of thousands of voxel signals captured across the entire brain. Even after rigorous voxel selection processes, approximately 10,000 voxels remain. On the other hand, fMRI data is heavily influenced by human behavior in response to visual stimuli, resulting in an extremely low signal-to-noise ratio.
>
> + The aforementioned hurdles complicate the decoding of precise and comprehensive ground truth movement from fMRI data in fMRI-to-video reconstruction. Nevertheless, through this endeavor, we present elaborated designed components and loss functions tailored for motion perception. These refinements enable NeuroClips to efficiently capture motion details and show impressive video reconstruction results. It is a valuable and motivating contribution to the fMRI decoding community. Further elaboration on these advancements is presented in the following responses.
>
> > **Question 1: No Model or Loss Function for Movement**
>
> ***NeuroClips is equipped with a specific model design and a tailored loss function, which guarantees its ability in capturing motion information.***
>
> * **Regarding Model Design.** The Perception Reconstructor (PR) module we designed can perceive movement (motion) information. In this work, we focus on capturing ***generic motion*** within videos by modeling the motion information from ***two perspectives***: perceiving the ***structured information within frames*** (pertaining to objects' shape, position, and orientation) and grasping the ***dynamic information across frame sequences*** (related to the movements of an object or the dynamics of the scene). To accomplish this, we introduce a spatio-temporal attention module within **Temporal Upsampling**, which is explicitly designed to decode spatiotemporal (structural-dynamic) details, i.e., motion cues, from fMRI data. We utilize the cues to guide the Video Diffusion model toward a more nuanced perception of motion dynamics, ultimately guaranteeing the motion-awareness of the video generation of NeuroClips. This is our meticulous design from the model perspective for capturing movement (motion) information.
>
> * **Regarding Loss Function.** The Mean Absolute Error (**`MAE`**) part within the loss function (**Eq. 1**) quantifies the disparity between the generated video and the groundtruth video frame-by-frame, facilitating the perception of generic motion within videos. Note that the perception embeddings of video frames, denoted as $\mathbf{E}_\mathcal{X}$, are aligned to the latent space of the Stable Diffusion Variational Autoencoder, which can be the equivalent of the pixel space at the frame level. Numerous recent video generation models have shown that the effective perception of generic motion can be achieved through two fundamental yet straightforward designs: **temporal attention mechanisms** and **frame-level loss functions** [1, 2]. Consequently, the design inspiration is embraced in this study.
>
> * **Regarding Experimental Evidences.** In **Figure 3**, the turtle's swimming direction, the airplane's orientation and flight direction, and the motorcycle rider's posture, along with their corresponding motion details, are **accurately reproduced**. If NeuroClips be exclusively crafted at a semantic level, it becomes evident that semantics alone are inadequate for capturing these motions. In addition, the exceptional accuracy of NeuroClips, as evidenced by the **pixel-level** and **ST-level** metrics in **Table 1**, further corroborates these visualization outcomes.
>
> * **Regarding Fine-grained Motion Decoding.** Currently, decoding fine-grained movements from fMRI data poses a **significant challenge** due to its **intricate and noisy nature**. Unlike text, fMRI data lacks straightforward cues such as motion-descriptive words, making the motion perception more complex. NeuroClips stands apart from previous text-to-video models that are capable of generating motions closely aligning with textual motion semantics. Note that NeuroClips has achieved significant progress in fMRI-to-video decoding tasks. Certainly, NeuroClips is by no means the final solution for fMRI-to-video decoding tasks; undoubtedly, superior solutions will emerge in the future. Looking ahead, there are **promising prospects** for enhancing the accuracy of decoding **fine-grained** movements from fMRI data in the future. This will be the focal point of our upcoming efforts.
>
>
> > **Question 2: Code Release**
>
> We promise to release the code at the earliest opportunity, following the acceptance of the paper. We appreciate and support open source because it helps more researchers contribute to the field and advances its development. This is an effective way to further enhance the significance and value of our research.
>
> **Reference**
>
> [1] Stable video diffusion: Scaling latent video diffusion models to large datasets, *Stability AI*
>
> [2] Align your latents: High-resolution video synthesis with latent diffusion models, *CVPR 2023*
>
> &nbsp;
> ***
> You have offered many constructive and valuable suggestions, making our submission more solid and complete. Once again, we sincerely express our best gratitude for your effort and time!
>
> Best wishes,
>
> All authors of Submission 351

---

> > ### Comment · Reviewer_85K8 · 2024-08-12
> >
> > Thanks for your well-constructive and persuasive review. I will raise the score.

---

> > > ### Author Response · Authors · 2024-08-12
> > >
> > > We greatly appreciate for your constructive feedback. We would like to express our sincere gratitude for your increased rating and further support towards our work! We hope that this paper achieves satisfactory results, not in vain of your efforts and suggestions.
> > >
> > > Once again, thank you, Reviewer 85K8.
> > >
> > > Best wishes,
> > >
> > > All authors of Submission 351

---

### Author Rebuttal · Authors · 2024-08-06

We sincerely thank all reviewers, AC, and SAC for their valuable time and selfless dedication. We are very pleased to see that the reviewers recognize the quality of our presentation (mcBd, oPVf), consider our experiments solid and extensive (mcBd, oPVf), and approve the novelty or soundness of NeuroClips (85K8, mcBd, oPVf). In particular, we are greatly encouraged by reviewer oPVf's recognition of the significance of our work. Meanwhile, we deeply value the reviewers' precious suggestions and questions, which we have addressed one by one in the rebuttal.

---

### Decision · Program_Chairs · 2024-09-25

**Decision:**

Accept (oral)

**Comment:**

This paper proposes NeuroClips, a framework that decodes high-fidelity and smooth video from fMRI. NeuroClips uses a semantics reconstructor for video keyframes to ensure semantic accuracy and consistency, and a perception reconstructor for capturing low-level perceptual details, ensuring video smoothness.
Additionally, the reviewers are convincingly shown that NeuroClips is equipped with a specific model design and a tailored loss function, which guarantees its ability in capturing motion information.
Questions on rate and resolution have been convincingly addressed.
Cross-subject modelling will be addressed in future work, which is fine.
Overall, the authors gave  detailed and convincing answers to all technical questions made by reviewers.
The contribution is both impressive and interesting, and thus certainly deserves a wide visibility within NeurIPS conference.